# Strategic attitude expressions as identity performance and identity creation in interaction
Caoimhe O'Reilly [1] ✉, Shane Mannion[2], Paul J. Maher[1], Elaine M. Smith [1], Pádraig MacCarron[1,2] & Michael Quayle [1,3]

We assess the strategic alignment of attitudes and the active construction of attitude-based identity across two studies. Study one assessed the twitter response (hashtags in English) to the war in Ukraine for five months after Russia's first invasion of Ukraine 2022 ($N = 8149$). Results demonstrated that individuals publicly expressed hashtags similar to others close to them in the followership network, showing their support for Ukraine and condemnation of the Russian invasion in qualitatively different ways. Study two was a preregistered Prolific experiment with geographical European participants ran in September, 2022 ($N = 1368$). Results demonstrated that attitude interaction with ingroup members motivated interactants towards attitude alignment, and attitude alignment strengthened the identification that motivated the alignment in the first place. Results suggest that attitude expression is performative and constrained by one's group relationship with one's audience and the definition of social identity can be constrained by opinion-based identity performance.

In periods of social change, especially those involving situations of threat, new groups based on new dimensions often emerge and it is often necessary to form new attitudes in response to novel contexts. In such periods of uncertainty, our ingroups often serve to inform us of the most appropriate norms, behaviours, and attitudes[1]. In February of 2022, the war in Ukraine escalated dramatically when Russia invaded Ukraine. This invasion sparked international outrage and condemnation, much of which was in the form of online attitude sharing. In the current research, we assess the co-ordination and alignment of online attitudes in relation to this novel context; the war in Ukraine. In our first study, we assess the online twitter response to the war in Ukraine to determine whether people tended to share similar attitudes as others in their follower network. We tracked tweets in relation to the war in Ukraine from the beginning of the war (February, 2022) and for the following five months.

In a second study, we assessed attitude identity dynamics and the strategic, performative nature of attitude-based communication in a controlled, experimental context, once again assessing attitudes related to the war in Ukraine. Importantly, we use the term performance to reflect how 'the expression of social identity [or an identity-relevant attitude] in behaviour is affected by considerations relating to the nature of available audiences' in line with previous theorists (ref. 2, p. 29, 5, 4). This does not imply that attitude expressions are merely due to audience effects (cf. the power of the situation, conformity, social desirability, or inauthenticity)—rather that attitudes achieve identity functions in social interaction.

We propose that attitude expressors are sensitive to their audience, and performatively express attitudes to consolidate ingroup identity[2–9]. To this end, we suggest that interaction with ingroup members should motivate people towards attitude alignment. Beyond this, we assert that the attitude-identity relationship is a dynamic, reciprocal process of influence[2,4,5]. That is, we suggest that the social act of attitude expression can affect the construction of social identification[10–15], and this social identification in turn can affect the attitudes that will be publicly expressed because of identity performance goals[2,4,5,16].

We know that awareness of attitude congruence through computer mediated communication can foster a sense of ingroup identification[17]. Our main hypothesis predicted that interaction with such ingroup members can influence future attitude expression as a result of strategic attitude [dis] alignment and identity performance[2,5,18]. Our secondary hypotheses predicted that, in a dynamic way, perceptions of similarity as a result of strategic and cumulative attitude alignment can strengthen the identification that motivated the alignment in the first place[10].

[1]Department of Psychology, University of Limerick, Limerick, Ireland. [2]Department of Maths, University of Limerick, Limerick, Ireland. [3]Department of Psychology, School of Applied Human Sciences, University of KwaZulu-Ng community into being on social network atal, Durban, South Africa. ✉e-mail: Caoimhe.Oreilly@ul.ie

## The social identity approach

In order to understand the theoretical rationale for the proposals in the current paper, we first must outline the theoretical model on which the rationale for these proposals is based: the social identity approach. The social identity approach[19,20] asserts that the implicit or explicit presence of others in a particular comparative context can imbue categories with social meaning. That is, 'the mere perception of common category membership' is sufficient and necessary for social identification and resultant group formation (ref. 21, p. 3). Beyond the effect of mere similarity[22], the intergroup context and awareness of a category one belongs to and another that one does not belong to, gives one's own category social meaning and thus it moves from being a mere category to being a social identity. Social identity is defined by Tajfel as 'the individual's knowledge that he [or she] belongs to certain social groups together with some emotional and value significance to him [or her] of this group membership' (ref. 21, p. 21). Social identification was defined by Turner as 'any social categorisation used by a person to define him-or herself and others' or 'the process whereby an individual internalises some form of social categorisation so that it becomes a component of the self concept, whether long-lasting or ephemeral' (ref. 20, p. 18). Self-categorisation theory explains that the process via which social categories become internalised as social identities is context dependent and varies (partly) as a function of the meta-contrast principle. The metacontrast principle stipulates that people cognitively structure social categories by adhering to elements in a given context that maximise relative differences between one's ingroup and one's outgroup and maximise the relative entitativity of one's ingroup[20].

In the late 90s and early 2000s there was a shift in social identity theorising towards recognising the performative, active nature of categorisation and social identity processes[6,8,9,23]. The notion of the purposeful deployment of categorisation was introduced, and theorists proposed that categories could be actively created, defined, and deployed[2,6,8,9,23,24]. That is, theorists began to recognise that categorisation is not something that just passively happens to people, but people have agency, ability, and motivation to be active agents in the construction of categories around which identities and groups revolve. Not only do 'social identities reflect social reality' but also 'social identities move people to create social reality' (ref. 9 p. 365). The social identity model of deindividuation effects (SIDE model) recognises the strategic nature of social identity[2,24] providing a theoretical framework for understanding how social identities can influence and create social reality via motivated identity performance.

## Performative expression of attitudes depending on audience

The SIDE model was originally developed to understand the effects of anonymity and identifiability especially within the realm of computer-mediated communication[24,25]. It has two main components, cognitive and strategic. The cognitive SIDE asserts that anonymity eliminates visual cues to identity, thus heightening the depersonalisation process and the likelihood for group identities to be salient. In 2007, the strategic SIDE was extended by Klein and colleagues to explain how public expression of identity can be performative and serve to consolidate, as well as construct, social identity[2]. Their theoretical assertion is that individuals' behaviour can be purposefully co-ordinated, motivated by a desire to be perceived as an ingroup member and to achieve ingroup acceptance[2,26,27]. The SIDE model provides a theoretical framework for understanding performative attitude expression. In line with the SIDE model we suggest that behavioural co-ordination can apply to attitudes, that attitude expression can be performative, and that people can be motivated towards attitude alignment when they express attitudes to their in-group members to publicly consolidate their identity.

This is consistent with the social identity approach at large. Self-categorisation theory asserts that, in an intergroup context, in line with the metacontrast principle, people should be motivated to be more similar to their ingroup to maximise the relative differences between themselves and their outgroup and to maximise the entitativity of their ingroup[20]. Similarly,

the referent informational influence theory[20,28] explains that ingroup members provide information about how to think and behave appropriately as a group member. As a result, attitudes perceived as normative of ingroup members are experienced as socially valid, accurate, and appropriate because ingroup members are relied on to provide information about appropriate social norms and are prioritized as a source of valid/correct information[29–32]. Thus, ingroup members should be motivated towards broad attitude alignment because (a) fellow ingroup members are perceived as having valid, normative, appropriate attitudes and (b) people in inter-group situations are motivated to be more similar to their ingroups and distinct from their outgroups[20,28].

Klein et al.[18] provide empirical support that attitude expression (specifically prejudice attitude expression) is constrained by one's audience and that attitude expressions can be strategically used to publicly consolidate one's identity[18]. In the current study, we suggest that the audience is a 'co-present interactional reality, itself susceptible to social influence' (ref. 5, p.21). Recognising this, we suggest that people in interaction can engage in a performative, active process of attitude co-ordination constrained by the identity relationship between interactants[4]. We suggest that attitudes that become normative are actively defined and constructed in interaction via a process of attitude prediction, desire to publicly express in-group consistent attitudes, and strategic attitude expression constrained by the ingroup/outgroup relationship between interactants[5].

In line with the social identity approach, we therefore propose that attitude expression can be an identity performance which is strategic and co-created depending on one's audience. Interaction with ingroup members should motivate interactors towards attitude alignment to overtly consolidate ingroup identity, because ingroup members provide information about which attitudes are correct/appropriate, and because ingroup members see themselves as more similar to other ingroup members in an intergroup context. Note that 'performativity' in this sense is agnostic about whether people are expressing their 'true' attitudes or not; the focus of analysis is on the function of attitude expression in achieving group identity and identification.

For such motivated attitude alignment to occur, people must first be interacting with ingroup members[2,20]. A plethora of previous research has found that salience of attitude congruence can foster ingroup identification[10–13,17,33–36]. In interaction, opinion-based identification is often the most contextually relevant means for categorising the self and others into groups[10–12,37]. This is especially the case when considering anonymous, computer mediated communication, where communication is often attitude based (likes, shares, tweets etc.) and visual identity cues are concealed[3,17]. Online, attitudes are often the only cues available from which to categorise others in group terms and under such conditions of relative anonymity we would expect depersonalisation and perceptions in group terms to be heightened[17,24]. Thus opinion-based identities may be an important and ubiquitous group type emerging from online communication[17,24].

In the current research we assert that, for identity-laden issues, computer mediated attitude congruence fosters ingroup identity with an audience whereas attitude incongruence fosters a sense of out-groupness[10]. Importantly, in the current research we are conceptualising categories and any resultant group identification, not as static objective groups (such as race, country of origin; see ref. 37 for a discussion), but as situational identities; ones that have been actively created within the interaction context, whose boundaries exist specifically and uniquely to that interactive context, and which can be updated with the introduction of new situationally relevant information (consistent with the self-categorisation theory's metacontrast principle, see also.[7,8,10,37–39]) Our main proposal is that interaction with people perceived as ingroup members will lead to alignment on attitudes, especially in contexts where attitudes are the primary means for achieving affiliation. Interaction with outgroup members may also motivate interactants towards attitude disalignment because people seek differentiation from outgroups, and outgroup members provide information about how not to behave[20].

## Performative attitude expression in the online context

Qualitative research has found that social identity can be actively [re]constructed and [re]defined on social media and that social media can be used as a tool to display and express one's social identity[40–42]. Furthermore, previous research has found that, when expressing attitudes online, people are aware of, and attentive to, their audience[43–45]. That is, the influence of the audience on communication translates to the online context[46]. Although people can comprehend the limitless nature of the social media audience, people nonetheless tend to behave as if the audience is more confined and imagine a particular audience when posting on social media[43,47].

In line with the social identity approach[19,20], in the online context (as in the offline context) people should be motivated to be similar to their online ingroups[20]. Posting an attitude online is a social act that goes beyond any similarity motivations however. That is, it is possible to be referently influenced by one's online ingroups without needing to post anything online[20]—the social act of sharing one's attitude online goes further than mere referent informational influence[20]. People may be motivated to publicly express attitudes online in order to consolidate and mobilise their identity[2,48]. People may look to others close to them in a social network and share similar attitudes to them because they are perceived to share an online social identity and because people are motivated to consolidate and mobilise this identity.

In line with our main proposal (that interaction with people perceived as ingroup members will lead to alignment on attitudes) we thus also explore whether people in online communities tend to express similar attitudes as those close to them in the social network. The purpose of this exploration is to determine whether patterns of attitudinal expression in the large scale online social media context suggest that people could be performatively expressing attitudes similar to their online groups to consolidate their online identities. We do not directly test the motivations behind these patterns, we simply explore and observe large-scale patterns of attitude expression within online communities on Twitter.

## Reciprocal relationship between performative attitude expression and opinion-based identity

As well as opinion-based identification influencing future motivated attitude alignment, we also expect this attitude alignment to strengthen the identification that motivated the alignment in the first place. Klein et al.[2] suggest that the relationship between context, identity, and identity performance is reciprocal. They explain that context influences one's salient social identity which in turn influences one's identity performance but that one's identity performance can also influence the definition of social identity and the context within which identity is occurring. Quayle applies a similar logic to attitudes specifically. He proposes that the relationship between attitudes and identity is dynamic, and recursive (4, see also 16 who apply this logic to computer-mediated communication).

In the current study, in line with Klein et al.[18] and Quayle[3], we suggest a dynamic reciprocal relationship between opinion-based identification and performative attitude expression. We propose that attitudes that are publicly expressed are contingent on the group nature of one's audience (whether the audience is perceived as ingroup or outgroup). Since people are attuned to how they will be positioned by their utterances, performative attitude expression can thus change the attitudes that enter the social world. As groups can be based on the attitudes expressed in context[11,12], performative attitude expression also has the potential to change social identification. That is, the definition of social identity can be constrained by opinion-based identity performance. Thus, the performative, strategic expression of attitudes can affect social reality itself[5].

Research has also found a cumulative attitude congruence effect, where attitude congruence on multiple attitudes strengthens identification[10]. We therefore expect that motivated attitude alignment can strengthen identification. Specifically, we suggest that the social act of attitude expression affects the construction of social identity, this social identification in turn effects the attitudes that will be publicly expressed (motivated attitude alignment), and the attitudes publicly expressed can, in turn, affect the

strength of identification. Importantly, we expect cumulative attitude congruence to strengthen identification for those interacting with opinion-based ingroup members but not those expressing attitudes to outgroups or privately expressing attitudes. We hypothesise that greater congruence will be related to greater identification for those expressing attitudes to ingroup members compared to outgroup members and those privately expressing attitudes. We also expect that attitude incongruence should weaken or even extinguish identification. Therefore, we also track change in identification as a result of changing attitude congruence. We hypothesise that change in congruence will predict change in identification over time (greater congruence will be related to greater identification) for those expressing attitudes to ingroup members.

## Methods

We run two studies, one an exploratory Twitter study, and the other a preregistered experiment (https://aspredicted.org/4dg7y.pdf). In our first study, we use real-world data to assess whether clusters of followers online tend to share similar attitudes to one another. Specifically, we gathered tweets from Twitter based on shared hashtags related to the Ukraine Russia crisis which escalated in 2022 with the Russian invasion of Ukraine. We captured a followership network, ran community detection algorithms to identify groups, and assessed the most popular hashtags shared by each follower community to visualise how attitude alignment occurs in the wild.

While study one uses large-scale real-world data to descriptively explore attitude alignment, in study two, we use controlled, interactive, online experimental methods to quantitatively explore motivated attitude alignment. Most experimental research designs are unable to capture recursive processes. Our experimental design allows us to capture the dynamic, co-occurring process where performative attitude expression shapes identity and where identity shapes performative attitude expression. We suggest that awareness of attitude congruence through computer mediated communication can foster identification[11,12,17] this identification influences subsequent attitude expression as a result of strategic attitude alignment[2,18]; and perceptions of similarity resulting from cumulative attitude alignment can strengthen identification[10]. We track change in identification over time as a result of changing attitude congruence to determine whether cumulative attitude congruence strengthens identification only when expressing attitudes to an ingroup audience. This design attempts to capture the dynamic and reciprocal nature of situated identity performance.

### Study one method

As outlined in the introduction, the social identity performance model asserts that people are motivated to consolidate their identity publicly. We suggest that one way people can achieve this is by publicly expressing attitudes that align with the attitudes of our ingroup members. Public attitude expression is ubiquitous in the online context. The main purpose of study one is to observe large-scale patterns of attitude sharing in 'real world' online contexts (as opposed to the more constrained experimental settings) and to explore how patterns of attitudinal expression might relate to the attitudes expressed by others close to oneself in online followership networks. In exploratory analysis, we assess whether individuals tend to publicly share attitudes (in the form of tweets) that are similar to others in their followership networks on Twitter. To do this, we tracked online expressions of attitudes in the form of hashtags on Twitter that related to the war in Ukraine for the five months following Russia's invasion of Ukraine (February, 2022).

Data were collected using the Twitter academic API specifically for the purpose of the current paper. We scraped Twitter for tweets in English containing hashtags relating to the Russia/Ukraine war, starting from the beginning of the invasion (24/02/22) and ending the day we commenced gathering data (28/06/22). Study one was not preregistered. Data and analytic code can be found here (https://osf.io/w8shc/?view_only= 34d38b51dc8a4691bf4fec2dbe3cde66).

## Participants and procedure

We searched Twitter to identify popular hashtags related to the ongoing invasion of Ukraine, as well as hashtags that were 'trending' at the time (see Supplementary Note 2 for full list of hashtags), attempting to identify as many dimensions of the online discourse as possible. Participants were included in the dataset if they had tweeted any of the hashtags on the hashtag list between February and June 2022 (from Russia's invasion of Ukraine and for five months after this). This yielded 5,178,508 tweets from 1,053,860 users. We then examined the frequencies of hashtag expressions for each tweet and chose six hashtags that had high tweet or retweet frequencies. Our final sample consisted of a subset of users who had tweeted one of these hashtags (Supplementary Note 4). We then created a dataset of all tweets in the original dataset from users who were in our final sample. Next, we gathered the follower list for each of these users, discarding the data for any user whose account information was unavailable through the Twitter API (e.g., restricted, banned, or deleted accounts). We also discarded any user in the follower lists for whom we had no tweets. This left us with 8149 users. We then built a followership network of these users, where users were linked to every user they were followed by, if that user had also tweeted or retweeted one of the hashtags.

The followership network of Twitter users is a directed network (i.e., there is a directionality associated with followership, unlike, for example, Facebook friendships) We thus employed the Infomap community detection algorithm to the directed version of the network[49]. The Infomap algorithm identifies communities based on how information may flow through the network, taking into account the directionality of the followership edges[50]. We then identified the five most common hashtags shared in each community detected by the algorithm. We excluded several hashtags such as #Ukraine, #Russia (Supplementary Note 3) because these were hashtags used by all communities and therefore not that informative of differences between the communities. That is, these hashtags did little more than signify that people were discussing the war in Ukraine crisis at large but were not informative of any qualitative nuances in the discourse. Using AFINN sentiment lexicon[51], we obtained a sentiment score from the tweets of each user.

## Statistics and reproducibility

The alpha level for all tests was .05 and all tests were two-tailed. Parametric tests were run on normal data and non-parametric tests were run when normality was violated. Data was analysed using SPSS version 26, R, and Python. Sufficient detail is included to enable reproducibility of this study (including open code and open data), although the particular context in which the Twitter data was collected will of course, not be reproducible (see also discussion section).

Using the Infomap community detection method we detected 611 communities, of which seven had more than 100 users (see Fig. 1). We then assessed the frequencies of the five most common hashtags for each of these seven communities. Supplementary note 5 represents the frequencies of people who shared each of these hashtags for all communities. It should be noted that in the communities containing the hashtag '#sanctionskill' (the blue community in both Figs. 1 and 2), many of the tweets were referring to sanctions against other countries, not necessarily Russia. However, as we can see in the network diagram (Fig. 1), this community is very densely connected to the rest of the network and so we did not remove these users.

We then compared the distribution of each hashtag across communities statistically using a significant chi-square goodness of fit statistic. We also ran individual ANOVA's to assess the frequency of individuals' hashtag use across all communities.

On top of the aforementioned aggregate level analysis, we also conducted individual level analysis. Using AFINN sentiment lexicon[51], we obtained a sentiment score from the tweets of each user and compared the average sentiment across the three largest communities using a one-way ANOVA.

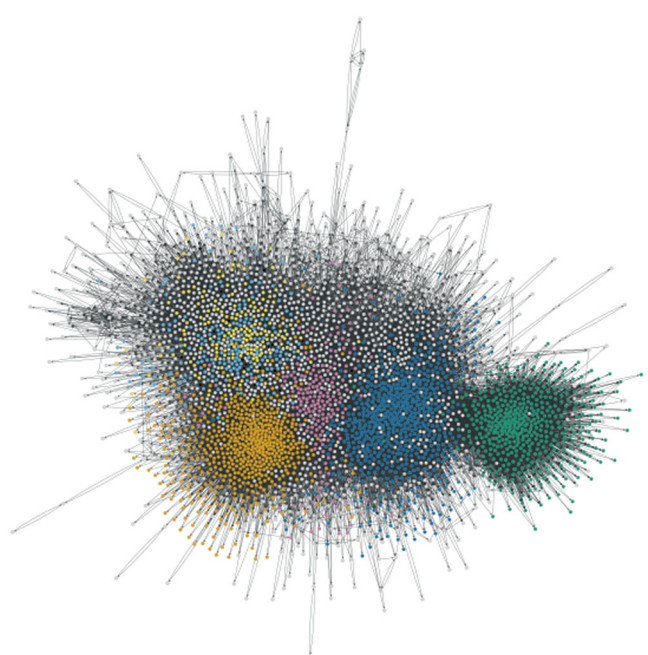

**Fig. 1 | Communities detected via the directed version of the followership network using the Infomap algorithm.** Community 0 is represented with bluish green nodes. Community 1 is represented with orange nodes. Community 2 is represented with blue nodes. Community 3 is represented with yellow nodes. Community 4 is represented with redish purple nodes. Community 5 is represented with sky blue nodes. Community 6 is represented with black nodes.

## Study two method

Study one allowed us to visualise real-world large-scale attitude expression patterns. This has the advantage of capturing real-world behaviour, however, this data is descriptive and, although it seems that attitude alignment occurs when publicly expressing attitudes to our followers, if we are to assert that attitude alignment strategically occurs when expressing attitudes to ingroup members it is necessary to test this in a more stringent, controlled environment. This is the purpose of study two.

As outlined in the introduction, in (implicit or explicit) intergroup contexts, people are motivated to be similar to their ingroups to maximise intergroup differences and intragroup entitativity (metacontrast theory,[20]) and because ingroups are perceived to hold appropriate norms and accurate information (referent informational influence theory,[20]). Furthermore, people are motivated to overtly display their identities via identity performance to consolidate and mobilise these identities (social identity performance model,[2]) In study two, we therefore propose that ingroup members should be motivated towards attitude alignment in line with the metacontrast theory and the referent informational influence theory. We also propose that performative attitude expression (in the form of attitude alignment with ingroup members) is one way that people can perform their identities in line with the social identity performance model.

Our main preregistered hypothesis (hypothesis one) predicted that attitude alignment would be greater in the ingroup experimental condition compared to the control condition and the out-group experimental condition. Our secondary hypotheses predicted that, hypothesis two: condition (ingroup vs outgroup vs control) would moderate the effect of congruence on identification. We expected congruence to predict identification better in the experimental (ingroup) condition, versus the outgroup and control, conditions. Hypothesis three predicted that change in congruence would predict change in identification over time (greater congruence would be related to greater identification) for the ingroup experimental condition. We expected congruence to predict identification better in the ingroup experimental, versus the outgroup and control conditions.

**Fig. 2 | Balloon graph graphical matrix of hashtag use for each community.** The dots in each cell represent the relative magnitude of the corresponding component i.e., larger dots represent greater frequencies.

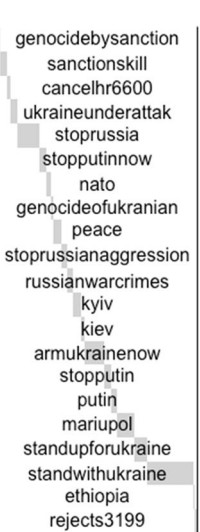

For the purpose of study two, it is important to distinguish between circumstantial attitude alignment and motivated attitude alignment. Of course, people who hold certain attitudes will be likely to hold other, related attitudes[52]. For example, if a person believes that their country of residence should give money towards Ukraine forces, they may also believe that their country of residence should give arms/weapons to Ukrainian forces because these attitudes are qualitatively similar. This is circumstantial attitude alignment—alignment that occurs because certain attitudes are interconnected and more likely to be co-held. Separate from this is motivated attitude alignment, which is co-ordinated, performative, strategic attitude alignment and which we suggest occurs when interacting in opinion-based intergroup terms. In study two, we compare circumstantial attitude [dis] alignment (the control group) to strategic attitude [dis]alignment (the experimental group) to ensure that any results found do not simply reflect patterns of qualitatively interlinked attitudes. Participants answer attitudes about the current crisis in Ukraine (2022–23).

In the experimental condition, participants share their attitude answers. In the control condition participants answer attitudes privately. In this way, we can compare motivated alignment occurring as a result of attitude expression and awareness of the intergroup context of one's audience (experimental group) to circumstantial alignment occurring as a result of holding qualitatively interrelated attitudes (control group). We compare circumstantial alignment to motivated alignment to assess whether interactants strategically align their attitudes depending on their group relationship with their audience in public attitude expression. We also compare opinion-based congruence to arbitrary non-opinion-based congruence to ascertain whether attitude congruence, rather than mere similarity, strengthens identification. We track change in identification over time as a result of changing congruence to assess how self-categories can be updated in context with the introduction of new, relevant information, and how attitude alignment can strengthen identification, whereas attitude disalignment can weaken it.

The study design, hypotheses, sample size, exclusion criteria and analyses were preregistered on the 12th of September 2022, prior to data collection (https://aspredicted.org/4dg7y.pdf). All analyses described in the preregistration are reported in the article or supplementary materials. All data, study materials, and analytic code are openly available here (https://osf.io/w8shc/?view_only=34d38b51dc8a4691bf4fec2dbe3cde66).

The study received ethical approval from the University of Limerick Committee (19_06_2019). All research is conducted ethically, results are reported honestly, and the submitted work is original and not plagiarised. On September 19th, 2022, data were collected via Prolific (version 2022) using Qualtrics (Version 2022). Prior to data collection, individuals who had participated in our previous studies were precluded (to avoid practice effects). The survey was made available to individuals who self-declared that their first language was English, who had a prolific approval rate greater than 94, who had a minimum of 50 prior survey submissions, and who were currently located in Europe (Supplementary Note 1). All participants who were included first gave informed consent.

Prolific samples are susceptible to several biases including first come first serve response bias, WEIRD bias, selection bias, and reward per hour bias (see ref. 53 for more detail). Nonetheless, crowdsourced data has been found to be of high quality[54] and Prolific data has been found to have higher quality of data than other crowd sourcing platforms[55]. It allows us to gather data from wide geographical areas, and data seems to function similarly well to laboratory studies[56].

## Participants

After preregistered exclusions (those who did not give consent ($N = 1$), who progressed less than 95% in the study ($N = 0$), who had less than two people in their dyads ($N = 27$), and who give the incorrect response to 'What is your letter?' ($N = 9$)), there were 1368 participants ranging in age from 18 to 75 ($M = 36.07$, SD = 12.27). Of these, 722 self-identified as female, 639 as male and 7 as non-binary. All participants were paid £1.80.

We conducted a sensitivity power analysis[57] with Anova (fixed effects one way) as the statistical test. Based on $N = 1368$, with alpha significance criterion .05 two-tailed and a standard power criterion of 80%, the analysis had power to detect an effect size of $f = 0.08$ (equivalent to $d = 0.16$).

After preregistered exclusions there were 432 people in the control group and 936 in the experimental group, of which 502 were in the experimental ingroup condition and 434 in the experimental outgroup condition.

## Materials

The exact survey can be found in the supplementary materials. All measures were assessed using distinct samples but identification was measured repeatedly at 11 timepoints.

Participants began by reading an information sheet which outlined a cover story explaining that the survey was about memory and attention. Next, they answered questions pertaining to informed consent, gender (male, female, non-binary), and age. Information on race/ethnicity was not collected.

All participants were asked 11 attitudes pertaining to their opinions about the Ukraine crisis (2022, 2023). Responses were binary (*yes/no; agree/ disagree*). Table 1 outlines all attitudes asked and percentage who agreed and disagreed.

**Table 1 | Stimulus Questions asked to all participants in study two**

| Attitude | % Agree (total sample) |
|---|---|
| Do you think your country of origin should extend unlimited welcome to as many refugees as need safe refuge? | 56.1 |
| Many companies have withdrawn from Russia. Some argue that it is necessary to destabilize Russia and undermines Putin's position and control. Others argue that this leaves innocent civilians without employment or access to necessary resources. Do you think that companies should withdraw from Russia? | 83.9 |
| Do you think NATO's expansion had a strong influence on Russia's invasion of Ukraine? | 59 |
| Do you think that the majority of the news you consume about the war is unbiased, accurate and fact checked? | 35.3 |
| Do you think your country of residence should give money towards Ukrainian forces? | 73.8 |
| Do you think your country of residence should give arms/weapons to Ukrainian forces? | 67.8 |
| Do you think Ukraine should be offered entrance into NATO? | 79.6 |
| Do you think your country of residence should send troops to fight alongside Ukrainian forces? | 27.5 |
| Do you think your country of residence should refuse to engage with Russia for as long as Putin remains president? | 77.9 |
| Social media sites such as Instagram and Facebook made exceptions to their policy against inciting violence, so long as the posts represented political expression against Russian forces invading Ukraine. Social media sites made the right decision to allow these posts inciting violence: | 27.3 |
| The current situation is that western Europe needs to import 75% of its oil and 50% of its gas, and Russia fulfils most of this requirement. Do you think that your countries should refuse to buy oil/gas from Russia, knowing that this may lead to inflation in fuel prices or oil/gas shortages? | 68.9 |

Participants in the control group were also asked 11 arbitrary information questions. An example is 'Is the third item in your prolific identification a letter or a number?' Responses were binary (*Letter/Number*).

Identification questions were phrased in group terms, for example 'I identify with the group my partner is representative of'. Identification was measured at eleven different time points (after each exposure/feedback round) with three items 'I identify with the group my partner is representative of'[58], 'I feel solidarity with the group my partner is representative of', 'I feel strong ties with the group my partner is representative of'[59]. All responses ranged from 1 *strongly disagree* to 7 *strongly agree*.

The current study has both an experimental and quasi experimental design. The experimental manipulation separated participants into the experimental condition who observed each other's attitude answers, and the control condition who observed each other's answers to the arbitrary non-attitude information questions. The quasi-experimental element involved further subdivision of the control and experimental groups based on their attitude congruence on the first attitude, into; experimental ingroup (those who are aware they have attitude congruence on the first attitude); experimental outgroup (those who are aware they have attitude incongruence on the first attitude); control ingroup (those who had attitude congruence on the first attitude but were unaware of this congruence); and control outgroup (those who had attitude incongruence on the first attitude but were unaware of this incongruence). Participants were not explicitly told they had been categorised into a group.

Congruence was a constructed variable based on whether a dyad answered the attitude (experimental group) or the arbitrary information question (control group) in the same way. If they answered the same, they got a score of one, if they answered differently, they got a score of zero. Total congruence was a sum of all 11 congruence scores thus, a total score of zero meant the two participants did not answer any question in the same way, a score of 11 meant that they answered all 11 questions in the same way.

Attitude congruence for the experimental groups were the same as the congruence variable described above. For the control groups, if both members of the dyad had the same attitude answer they got a score of one, and if they had different answers they got a score of zero. Importantly, the control group participants were unaware of their partner's attitude answer. Attitude alignment was a sum of all 11 attitude congruence scores.

As well as assessing attitude alignment within dyads, we also assessed bipartite attitude alignment within conditions. To do this, we created bipartite graphs which connected users via attitude agreement[60,61]. In these bipartite graphs there were two types of nodes, one representing attitudes, and the other representing each participant. Participant nodes were linked to attitude nodes via positive edges if a participant agreed with an attitude, or via negative edges if a participant disagreed with an attitude. Next, following

MacCarron et al.[60], network projections were created which connected each participant to other participants based on how much attitude congruence they had with each other (60:61). Participants would be very close in the network if they had lots of the same attitudes, and further away if they had lots of different attitudes. We created four network projections (bipartite graphs), one for each condition. We then compared the attitude alignment across conditions statistically. After creating the networks where the edges represent shared agreement of participants, we computed the clustering coefficient and the average path length. The clustering coefficient is the fraction of closed triangles in the network, if a node is connected to two neighbours, it gives the probability those neighbours are also connected (see ref. 62 for further explanation). The average path length represents the mean shortest number of steps between a pair of participants. In social networks this is related to the idea of six degrees of separation which hypothesises that the mean number of steps between any two people is six.

**Procedure**

The current study involved computer-mediated interaction. Participants participated in pairs, into which they were randomly matched via Qualtrics programming (following[63]) Participants read an information sheet, gave informed consent, and answered demographic questions. We recognise that individuals interacting in dyads can perceive themselves to be engaged in intergroup interaction, just as individuals interacting in groups can perceive themselves as individuals rather than as group members[64,65]. To maximise the likelihood of intergroup rather than interpersonal interaction participants were told 'You have been matched with another participant. This participant has been chosen because they are representative of a particular sample of people'. Next, they engaged in a round of instant messaging for 180s with their interactive partner, the purpose of which was to help participants understand they were interacting with a real person (which they were).

Participants then gave an attitude about the Ukraine crisis (2022/2023). Participants in the control group also answered an arbitrary information question. Here the experimental manipulation occurred. Participants in the experimental group observed each other's answers to the Ukraine attitude, whereas participants in the control group observed each other's answers to the arbitrary information question. Thus, the experimental group shared attitude answers (which allowed them to observe whether they had attitude congruence or not), while the control group privately answered the same attitudes but shared arbitrary information (which allowed them to observe whether they had congruence on the arbitrary information or not). This allowed us to determine whether any attitude alignment and identification effects found were as a result of similarity, generally, or whether they were a result of *attitude* similarity in particular. All participants answered attitude

one, and participants in the control group additionally answered arbitrary information question one. Following this, all participants were asked their identification with the sample that the other participant represented. Participants then engaged in ten more rounds of information sharing and identification reporting. Finally, participants answered items on superordinate opinion-based social identification, activism intentions, and their country of birth and current residence, before being debriefed.

## Statistics and reproducibility

The alpha level for all tests was 0.05 and all tests were two-tailed. Parametric tests were run on normal data and non-parametric tests were run when normality was violated. Data was analysed using SPSS version 26, R, and Python. Sufficient detail is included to enable reproducibility of experiments, including open materials open analytic code, and open data.

As a manipulation check, to assess whether the experimental ingroup had higher identification after the experimental manipulation than the experimental outgroup, we conducted a one-way ANOVA, with condition as the independent variable and identification at time one as the dependent variable.

To test hypothesis one we conducted a one-way ANOVA with Bonferroni post hoc tests, with condition as the independent variable and total attitude congruence as the dependent variable. We deviated slightly from the preregistration by subdividing the control group in the same way as the experimental group. Analysis without this subdivision is available in the supplementary materials but we caution that this analysis fails to control for circumstantial attitude alignment.

As well as assessing attitude alignment within dyads, we also assessed overall bipartite attitude alignment within each condition via generating bipartite graphs which connect participants via the attitudes they share (Fig. 3)[60,61]. Firstly, we connected participants who had congruence on all 11 items. We then lowered the agreement threshold until a majority of participants were in a giant connected component. The threshold was nine items. We then assessed two network quantities; path lengths and clustering coefficients. The path length assesses how 'coherent' the attitude space is, so shorter path lengths (shorter distances between nodes) represent higher attitude congruence, and longer path lengths (longer distances between nodes) represent lower attitude congruence. We then assessed whether the average path lengths differed across condition via one-way ANOVA. The clustering coefficient is robust to unequal sample sizes and assesses how 'clustered' a group is or how many triangles there are. If one's neighbours in the attitude network all agree with each other, this will result higher clustering, if not then we would observe lower clustering. As the data were not normally distributed, we assessed differences in clustering across condition via a Kruskal-Wallis.

Hypothesis two predicted that condition (ingroup vs outgroup vs control) would moderate the effect of congruence on identification. We expected congruence to predict identification better in the experimental (ingroup), versus the outgroup and control, condition. We conducted a linear model with fixed effects of total congruence, condition, and their interaction on total identification averaged across the rounds.

Hypothesis three predicted that change in congruence would predict change in identification over time (greater congruence will be related to greater identification) for the ingroup experimental condition. We expected congruence to predict identification better in the ingroup experimental, versus the outgroup and control conditions. In order to test this, a linear mixed model using maximum likelihood estimation was conducted with identification and congruence at each time point, nested within participants. A variable denoting the eleven rounds was included as a repeated effect. A random intercept was specified in order to account for individual variation in the tendency for identification. Fixed effects of congruence, condition, time, and their interactions were included. No random slopes were specified in order to retain a more parsimonious model. Importantly, we coded congruence as present[1] or not(0) at each point rather than using total congruence scores.

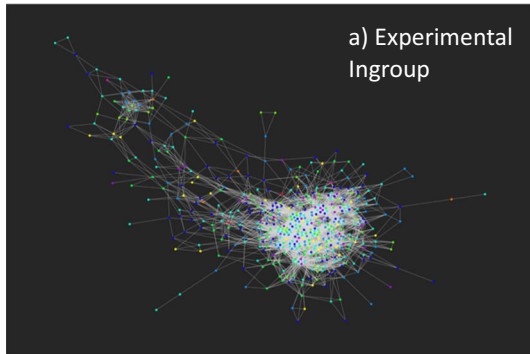

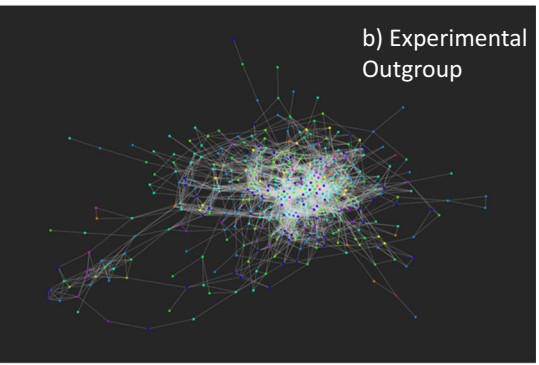

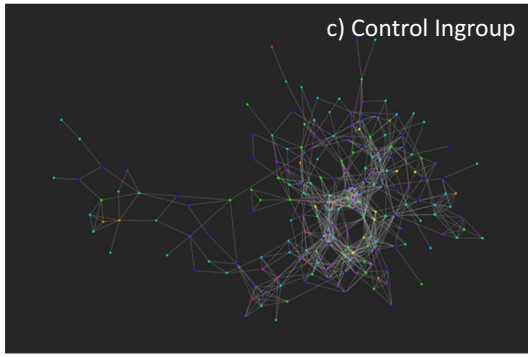

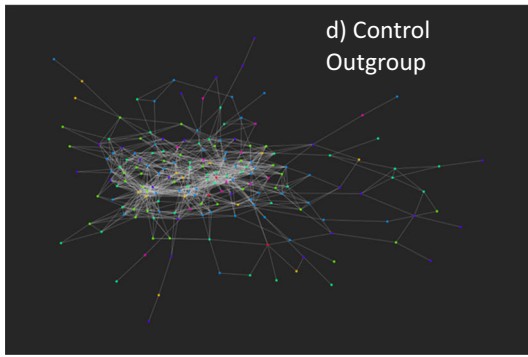

**Fig. 3 | Bipartite attitude alignment for each condition.** The network for the experimental ingroup condition is more dense with a lower mean path length. This suggests that there is greater attitude alignment in the experimental ingroup condition. The nodes are coloured based on their identification score represented by a rainbow spectrum An edge is created when the participants agree on X or more of the 11 items. **a** represents the experimental ingroup, **b** the experimental outgroup, **c** the control ingroup and **d** the control outgroup.

**Table 2 | Communities detected via the directed version of the followership network using the Infomap algorithm as well as the five most popular hashtags from each group**

| | Communities | | | | | | |
|---|---|---|---|---|---|---|---|
| | 0 | 1 | 2 | 3 | 4 | 5 | 6 |
| Size | 1398 | 907 | 969 | 198 | 382 | 285 | 772 |
| Colour | Bluish Green | Orange | Blue | Yellow | Reddish Purple | Sky Blue | Black |
| #genocidebysanctions | X | | | | | | |
| #sanctionskill | X | | | | | | |
| #cancelhr6600 | X | | | | | | |
| #ukraineunderattaÑk | | | X | | | | |
| #stoprussia | | X | | X | X | X | X |
| #stopputinnow | | X | | | | | |
| #nato | | | X | | | | |
| #genocideofukranians | | | | X | | | |
| #peace | | | X | | | | |
| #stoprussianaggression | | | | | | | X |
| #russianwarcrimes | | X | | | | | X |
| #kyiv | | | X | | | | |
| #kiev | | | X | | | | |
| #armukrainenow | | X | | X | X | | X |
| #stopputin | | | | | | X | X |
| #putin | | | | | | X | |
| #mariupol | | | | X | X | X | |
| #standupforukraine | | | | | X | | |
| #standwithukraine | | X | | X | X | X | X |
| #ethiopia | X | | | | | | |
| #rejects3199 | X | | | | | | |

An 'X' represents that a particular hashtag was one of the six highest hashtags tweeted for a particular community.

## Reporting summary

Further information on research design is available in the Nature Portfolio Reporting Summary linked to this article.

## Results

### Study one

Results demonstrated that the five most popular hashtags were not the same in each community (Table 2; Fig. 2). Descriptive results demonstrated that the algorithm detected three large communities each with different popular hashtags, with little to no overlap (Supplementary Note 5). The algorithm also detected several smaller communities with some overlap, but still with hashtags unique to their community.

The distribution of hashtags was not even across communities and hashtag use characterised these communities (Table 3). This demonstrated that the results depicted in Table 2 differed significantly from a random distribution. ANOVA's demonstrated that 20 out of 21 showed statistically significant differences (Table 4). Thus, followership communities on Twitter are characterised and clearly distinguished by the combination of hashtags that they publicly expressed. Taken together, these results all suggest that groups of followers tend to use similar hashtags depending on what 'group/community' they are part of.

A one-way ANOVA (see supplementary note 6, supplementary note 7) demonstrated that the average sentiment for individuals in the followership network was significantly different across communities (community zero: $M = -1.35$, $SD = 1.63$; community one: $M = -0.51$, $SD = 1.60$; community two: $M = -0.85$, $SD = 1.68$), $F(2, 4290) = 78.86$, $p < 0.001$ $\eta_P^2 = 0.035$, 95% CI 0.03, 0.05]). These results demonstrate that, on average, variation in sentiment is larger across communities than it is within communities

**Table 3 | Chi square goodness of fit statistics demonstrate that the distribution of hashtags was not even across communities**

| | Chi Square |
|---|---|
| #genocidebysanctions | $X^2 (1, N = 2252) = 2246.01, p < 0.001$ |
| #sanctionskill | $X^2 (3, N = 4349) = 9483.68, p < 0.001$ |
| #cancelhr6600 | $X^2 (1, N = 2818) = 2806.01, p < 0.001$ |
| #ukraineunderattaÑk | $X^2 (6, N = 4870) = 13952.07, p < 0.001$ |
| #stoprussia | $X^2 (6, N = 16105) = 15857.64, p < 0.001$ |
| #stopputinnow | $X^2 (6, N = 5067) = 4165.60, p < 0.001$ |
| #nato | $X^2 (6, N = 3658) = 9406.14, p < 0.001$ |
| #genocideofukranians | $X^2 (5, N = 1733) = 2083.06, p < 0.001$ |
| #peace | $X^2 (6, N = 5781) = 14688.15, p < 0.001$ |
| #stoprussianaggression | $X^2 (5, N = 3710) = 3158.85, p < 0.001$ |
| #russianwarcrimes | $X^2 (6, N = 4516) = 4369.46, p < 0.001$ |
| #kyiv | $X^2 (6, N = 6131) = 3951.83, p < 0.001$ |
| #kiev | $X^2 (5, N = 2350) = 10278.13, p < 0.001$ |
| #armukrainenow | $X^2 (6, N = 14367) = 16396.54, p < 0.001$ |
| #stopputin | $X^2 (6, N = 4898) = 4471.25, p < 0.001$ |
| #putin | $X^2 (6, N = 4282) = 1520.34, p < 0.001$ |
| #standupforukraine | $X^2 (5, N = 8567) = 21134.53, p < 0.001$ |
| #standwithukraine | $X^2 (6, N = 33719) = 22859.98, p < 0.001$ |
| #ethiopia | $X^2 (2, N = 1198) = 2380.07, p < 0.001$ |
| #rejects3199 | $X^2 (N=) =, p =$ |

**Table 4 | One way ANOVA's with community as the independent variable and each hashtag as the dependent variable demonstrate that the usage of each hashtag was different across communities**

| | ANOVA |
|---|---|
| #genocidebysanctions | $F_{(12, 5324)} = 38.56$, $p < 0.001$ $\eta_p^2 = 0.08$, 95% CI[0.06, 0.09] |
| #sanctionskill | $F_{(6, 4904)} = 54.87$, $p < 0.001$ $\eta_p^2 = 0.06$, 95% CI[0.05, 0.08] |
| #cancelhr6600 | $F_{(6, 4904)} = 56.47$, $p < 0.001$ $\eta_p^2 = 0.06$, 95% CI[0.05, 0.08] |
| #ukraineunderattaÑk | *NOT SIG* |
| #stoprussia | $F_{(6, 4904)} = 15.82$, $p < 0.001$ $\eta_p^2 = 0.02$, 95% CI[0.01, 0.03] |
| #stopputinnow | $F_{(6, 4904)} = 4.42$, $p < 0.001$ $\eta_p^2 = 0.01$, 95% CI[0.01, 0.01] |
| #nato | $F_{(6, 4904)} = 3.00$, $p < 0.001$ $\eta_p^2 = 0.01$, 95% CI[0.00, 0.01] |
| #peace | $F_{(6, 4904)} = 1.55$, $p < 0.001$ $\eta_p^2 = 0.01$, 95% CI[0.00, 0.01] |
| #stoprussianaggression | $F_{(6, 4904)} = 7.07$, $p < 0.001$ $\eta_p^2 = 0.01$, 95% CI[0.01, 0.01] |
| #russianwarcrimes | $F_{(6, 4904)} = 3.09$, $p < 0.001$ $\eta_p^2 = 0.01$, 95% CI[0.00, 0.01] |
| #kyiv | $F_{(6, 4904)} = 1.99$, $p < 0.001$ $\eta_p^2 = 0.01$, 95% CI[0.00, 0.01] |
| #kiev | $F_{(6, 4904)} = 1.60$, $p < 0.001$ $\eta_p^2 = 0.01$, 95% CI[0.00, 0.01] |
| #armukrainenow | $F_{(6, 4904)} = 22.54$, $p < 0.001$ $\eta_p^2 = 0.03$, 95% CI[0.02, 0.04] |
| #stopputin | $F_{(6, 4904)} = 8.86$, $p < 0.001$ $\eta_p^2 = 0.01$, 95% CI[0.01, 0.02] |
| #putin | $F_{(6, 4904)} = 110.09$, $p < 0.001$ $\eta_p^2 = 0.01$, 95% CI[0.01, 0.02] |
| #mariupol | $F_{(6, 4904)} = 3.93$, $p < 0.001$ $\eta_p^2 = 0.01$, 95% CI[0.01, 0.01] |
| #standupforukraine | $F_{(6, 4904)} = 3.66$, $p < 0.001$ $\eta_p^2 = 0.01$, 95% CI[0.01, 0.02] |
| #standwithukraine | $F_{(6, 4904)} = 47.20$, $p < 0.001$ $\eta_p^2 = 0.06$, 95% CI[0.04, 0.07] |
| #ethiopia | $F_{(6, 4904)} = 69.20$, $p < 0.001$ $\eta_p^2 = 0.08$, 95% CI[0.06, 0.09] |
| #genocideofukranians | $F_{(6, 4904)} = 4.15$, $p < 0.001$ $\eta_p^2 = 0.01$, 95% CI[0.01, 0.01] |
| #rejects3199 | $F_{(6, 4904)} = 49.08$, $p < 0.001$ $\eta_p^2 = 0.06$, 95% CI[0.04, 0.07] |

## Study two

**Manipulation check**. As expected, identification was significantly higher in the experimental ingroup condition ($M = 5.21$, $SD = 1.02$, 95% CI 5.12, 5.30), compared to the experimental outgroup condition ($M = 3.54$, $SD = 1.11$, 95% CI 3.43, 3.64]; $F_{(1, 935)} = 575.10$, $p < 0.001$, $\eta_p^2 = 0.38$, 95% CI [0.34, 0.42]).

**Main hypothesis: Hypothesis one**. Attitude alignment was highest in the experimental ingroup condition ($n = 502$), followed by the control ingroup ($n = 220$), the control outgroup ($n = 212$), and lowest in the experimental outgroup ($n = 434$) condition: $F_{(3, 1362)} = 37.44$, $p < 0.001$ $\eta_p^2 = 0.08$, 95% CI [0.05, 0.10] as shown in Table 5. Results when randomly selecting one person from each dyad were the same (Supplementary Note 8). As expected, the experimental ingroup condition had significantly higher attitude alignment than all other conditions (vs the experimental outgroup condition ($p < 0.001$; 95% CI [0.79, 1.37]); vs the control ingroup condition ($p = 0.02$; 95% CI [0.04, 0.76]); and vs the control outgroup condition ($p < 0.001$; 95% CI [0.59, 1.31]) indicating that publicly expressing attitudes to an ingroup member and seeing their attitudes leads to some kind of strategic motivation for attitude alignment. The null hypothesis was rejected. In our preregistration, we noted that we suspected that experimental outgroup attitude alignment would be lower than the control group. As suspected, Bonferroni comparisons demonstrated that the experimental outgroup condition had significantly lower attitude alignment than the control ingroup condition ($p < 0.001$; 95% CI [−1.05, −0.32]) suggesting that interaction with an outgroup member leads to strategic motivation for attitude disalignment.

**Bipartite attitude alignment**. Results demonstrated that the experimental outgroup has statistically significantly shorter path lengths ($M = 4.63$) than the experimental ingroup ($M = 4.67$, $p < 0.001$), $F_{(2, 1362)} = 12.33$, $p < 0.001$ $\eta_p^2 < 0.001$, 95% CI [<0.001, <0.001]). This does not support our preregistered hypothesis, however, path length analysis is sensitive to unequal sample sizes and there were 502 people in the experimental ingroup and 434 in the experimental outgroup. The longest path length possible was 15 and the shortest was 1. Such a small difference between means (.03), although statistically significant, because of the unequal sample sizes, is likely to reflect the path length's sensitivity to sample size rather than reflecting any meaningful difference. Clearly, more work is needed to develop network methods to assess psychologically relevant features, and real-life data which is inevitably susceptible to unequal sample sizes.

Results demonstrated that the experimental ingroup has statistically significantly higher clustering ($Md = 0.40$) than all other conditions (experimental outgroup: $Md = 0.35$, control ingroup: $Md = 0.36$, control outgroup: $Md = 0.36$); $H^3 = 25.37$, $p < 0.001$). The density of edges was higher in the experimental ingroup condition (.05) compared to the control condition (0.04). This indicates that attitude alignment is highest in the experimental ingroup.

**Hypothesis Two**. No statistically significant main effect of total congruence ($b = 0.02$, SE = 0.03, $t = 0.92$, $p = 0.36$, 95% CI [−0.03, 0.08]) was found. Condition was dummy-coded, with control group used as a comparison. There was a significant main effect of being in the experimental ingroup (vs control) ($b = −1.08$, SE = 0.24, $t = −4.48$, $p < 0.001$, 95% CI [−1.56, −0.61]) and the experimental outgroup (vs control) ($b = −1.13$, SE = 0.23, $t = −4.83$, $p < 0.001$, 95% CI [−1.58, −0.67]). There was a significant experimental ingroup (vs control) × congruence interaction ($b = 0.25$, SE = 0.04, $t = 6.86$, $p < 0.001$, 95% CI [0.18, 0.32]) and an experimental outgroup (vs control) × congruence interaction ($b = 0.18$, SE = 0.04, $t = 4.81$, $p < 0.001$, 95% CI [0.11, 0.26]). The relationship between congruence and identification was strongest in the ingroup, and stronger in the outgroup condition than in the control condition (see Fig. 4). Thus, hypothesis two was supported.

**Hypothesis three**. The linear mixed model revealed a significant main effect of congruence ($b = 0.14$, SE = 0.05, $t = 3.01$, $p = 0.003$ 95% CI [0.05, 0.23]) and time ($b = −0.02$, SE = 0.01, $t = −3.97$, $p < 0.001$, 95% CI

**Table 5 | Attitude alignment means and standard deviations for each condition**

| | Control group (private attitude expression) | Experimental group (public attitude expression) |
|---|---|---|
| Ingroup (begin with attitude congruence) | M = 6.76, SD = 1.69 | M = 7.16, SD = 1.72 |
| Outgroup (begin with attitude Incongruence) | M = 6.21, SD = 1.45 | M = 6.08, SD = 1.70 |

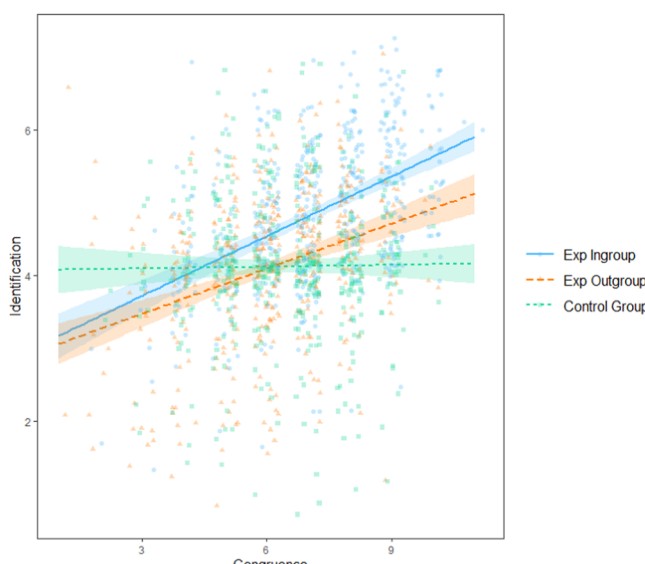

**Fig. 4 | The effect of total congruence on total identification for each condition.** Shaded regions represent 95% confidence intervals (*n* = 836). The blue solid line represents the experimental ingroup. The orange long-dashed line represents the experimental outgroup. The green short-dashed line represents the control group.

[−0.03, −0.01]). Condition was dummy-coded, with the control group used as a comparison. There was a significant main effect of being in the experimental ingroup (vs control) ($b = 0.29$, SE = 0.08, $t = 3.58$, $p < 0.001$, 95% CI [0.13, 0.45]), and the experimental outgroup (vs control) ($b = −0.67$, SE = 0.08, $t = −8.48$, $p < 0.001$, 95% CI [−0.83, −0.52]). There was a significant experimental ingroup (vs control) × congruence interaction ($b = 0.60$, SE = 0.07, $t = 9.04$, $p < 0.001$, 95% CI [0.47, 0.73]), and an experimental outgroup (vs control) × congruence interaction ($b = 0.83$, SE = 0.07, $t = 12.67$, $p < 0.001$, 95% CI [0.70, 0.96]). The effect of congruence was stronger in each experimental condition, versus the control condition. This is the effect of sharing attitudes within dyads.

The experimental ingroup (vs control) × time interaction was not statistically significant ($b = −0.00$, SE = 0.01, $t = −0.05$, $p = 0.97$, 95% CI [−0.02, 0.02]). The experimental outgroup (vs control) × time interaction was significant ($b = 0.03$, SE = 0.007, $t = 3.92$, $p < 0.001$, 95% CI [0.01, 0.04]). This indicates that the difference in identity between the control group and the outgroup changed over time.

The congruence by time interaction was not statistically significant ($b = 0.004$, SE = 0.01, $t = 0.63$, $p = 0.53$, 95% CI [−0.01, 0.02]). The experimental ingroup condition/time/congruence interaction was not statistically significant ($b = 0.01$, SE = 0.01, $t = 1.02$, $p = 0.31$, 95% CI [−0.01, 0.03]), nor was the experimental outgroup condition/time/congruence interaction ($b = 0.02$, SE = 0.01, $t = 0.71$, $p = 0.48$, 95% CI [−0.01, .03]). Probing these results with analysis split by condition revealed that congruence predicted identification over time for the experimental ingroup ($b = 0.02$, SE = 0.007, $t = 2.11$, $p = 0.04$, 95% CI [0.001, 0.03]) but not for the experimental outgroup ($b = 0.01$, *SE* = 0.008, $t = 1.40$, $p = 0.16$, 95% CI [−0.01, 0.03]) or the control group ($b = 0.005$, SE = 0.005, $t = 0.90$, $p = 0.37$, 95% CI [−0.01, 0.02]). Figure 5 illustrates this effect by showing how congruence accumulated over time. It is clear from the graph that in the ingroup condition in particular, this accumulation corresponded with an increase in identification. These results suggest that our analysis did not have sufficient power to

detect a three-way interaction but the split group analysis and Fig. 5 clearly demonstrate support for our hypothesis. Figure 6 demonstrates that congruence (vs incongruence) was associated with higher identification for all conditions, but the highest identification related to attitude congruence was in the experimental ingroup condition, and the lowest identification associated with attitude incongruence was in the experimental outgroup condition.

As can be seen in Fig. 6 (see also Supplementary Note 9), at every time point participants in the experimental ingroup with congruence had significantly higher identification than all other conditions. With the exception of two comparisons, the experimental outgroup with incongruence had significantly lower identification than all other conditions. Thus, having attitude incongruence had the strongest identification weakening effects for those interacting with outgroup members.

Participants in the experimental ingroup with incongruence had higher identification than participants in the experimental outgroup with incongruence with the exception of time 7. Participants in the experimental outgroup with incongruence had lower identification than participants in the experimental ingroup with incongruence at each time point.

## Discussion
The studies in the current paper were carried out in the context of the ongoing war in Ukraine to assess attitude identity dynamics as they emerge in novel contexts and the strategic, performative nature of attitude-based communication. We assessed whether attitudes in the real-world online context are aligned with attitudes of those in our followership network by assessing the most popular hashtags shared by follower communities on twitter in the first five months of the war in Ukraine (2022). We found that individuals publicly expressed hashtags that were similar to others close to them in the follower network, thus showing their support for Ukraine and condemnation of the Russian invasion but in qualitatively different ways. These results imply that public expression of attitudes online is influenced by, or at least associated with, one's followership community.

These results demonstrate that individuals tend to publicly express hashtags that others in their follower communities also express. When looking at tweets about the Ukraine crisis, several communities expressed their support for Ukraine (#standwithukraine) but in qualitatively different ways. Some expressed support by tweeting attitudes relating to arming Ukraine (#armukrainenow), some focused on raising awareness about Russian war crimes (#russianwarcrimes) or raising awareness about Ukrainian casualties (#genocideofukrainians), others advocated to sanction Russia (#boycottrussianoilnow), or to stop Russia or Putin (#stoprussia, #stopputin) whilst others focused on advocating for peace or Ukraine's NATO status (#peace, #nato). Importantly, the public expression of one's support for Ukraine was similar to those closest to oneself in the followership community. This implies that public expression of attitudes on twitter is affected by one's follower network—as different clusters of followers publicly expressed tweets that were qualitatively similar to others close to them in the follower network. According to the social identity approach at large, and the social identity performance model, people should be motivated to share attitudes that are similar to their ingroups in order to consolidate and mobilise their identity. The attitude expression patterns that we have observed in the Twitter context in the 5 month aftermath of Russia's first invasion of Ukraine are consistent with this interpretation as people tended to share attitudes similar to those closest to them in the twitter followership network. Thus, using large-scale, 'real world', naturalistic data we have observed patterns of attitude sharing consistent with the social identity performance model implying that people's public attitude

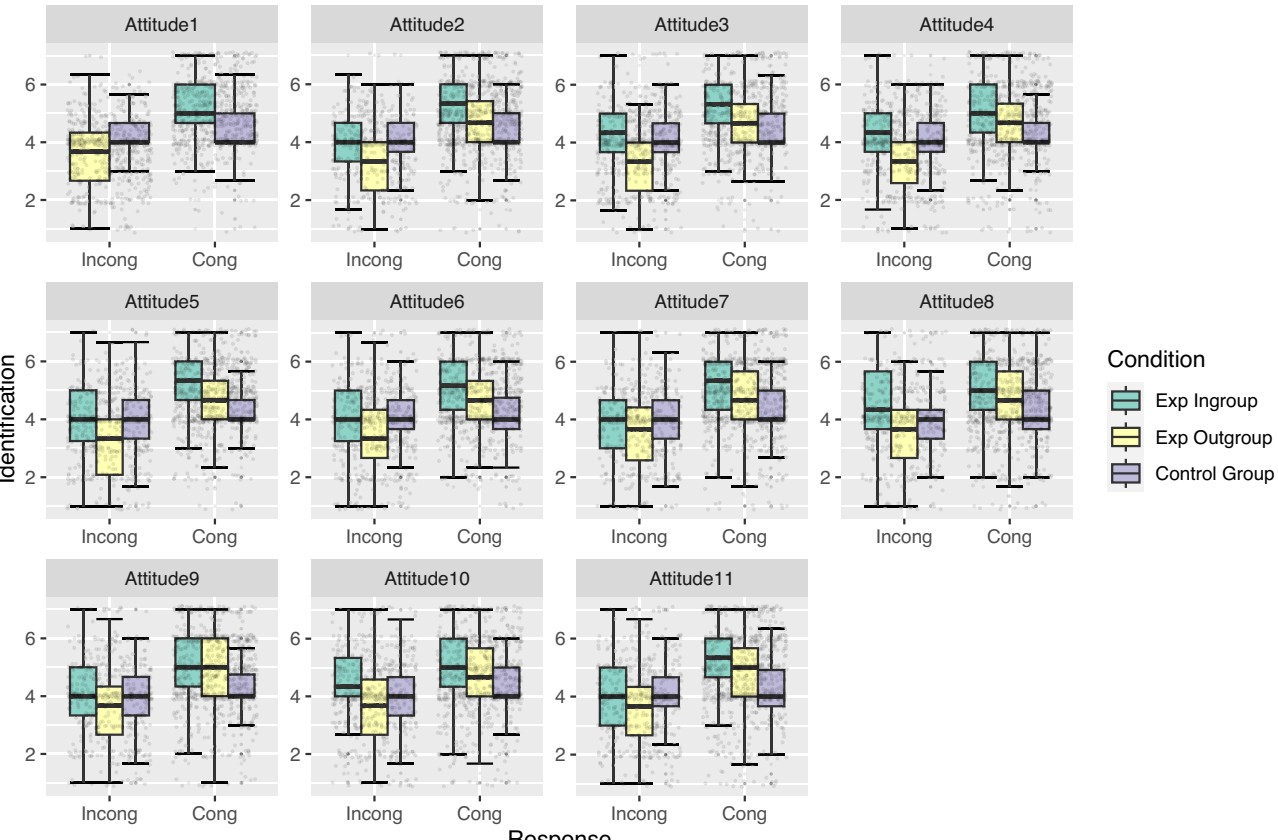

**Fig. 5 | The effect of cumulative congruence on identification at each item/time point for the experimental ingroup, outgroup, and control conditions.** Higher values on the graph are represented by increasingly saturated purple nodes. Green represents the experimental ingroup. Yellow represents the experimental outgroup. Purple represents the control group.

expression is performative and constrained by identity motivations. Nonetheless, we have not directly measured or manipulated identification thus, although the patterns are consistent with the social identity performance model, we can only speculate why these attitudinal expression patterns have occurred.

In a more controlled experiment, we then assessed whether public expression of attitudes can be considered as identity performance constrained by the identity relationship of expressor and audience. In particular, we assessed whether people were motivated towards attitude alignment when interacting with ingroup members. We also assessed the dynamic relationship between attitudes and identities to determine whether motivated attitude alignment could strengthen the identification which motivated the attitude alignment in the first place. We used an interactive experimental design allowing us to observe the attitude identity dynamic as it changed over time in a controlled environment, a process difficult to capture in the experimental context, but crucial if we are to understand the active processes involved in social identity and social reality development over time[7,8].

We found support for our preregistered predictions. Firstly, we found that public expression of attitudes was contingent on the identity relationship between interactants, implying that attitude expression is performative and constrained by, or tailored to, one's audience. Interaction with ingroup members motivated interactors towards attitude alignment whereas interaction with outgroup members motivated interactors towards attitude disalignment. Importantly, this motivated alignment was different to circumstantial alignment patterns found in the control group who privately expressed their attitudes. On top of finding higher attitude alignment within dyads in the experimental group, we found higher attitude alignment between individuals in the experimental ingroup more so than mere

situational alignment (control ingroup condition) or interaction with an outgroup member. These findings suggest that interaction with an ingroup member can lead to alignment of an entire opinion space. Nonetheless, we were unable to robustly assess path length comparisons across groups due to unequal sample sizes thus caution is needed when interpreting these results and future research should assess this using equal samples.

That the entire ingroup aligned their attitudes after interaction with only one ingroup member, could potentially reflect individual ingroup members' strategic tailoring of expressed attitudes to be consistent with the attitude they perceive to most prototypical. That is, bipartite attitude alignment may be representing a shift towards perceived ingroup prototypical attitudes. In our study, people began with either congruence or incongruence on their attitude about whether their country of origin should extend unlimited welcome to refugees. Perhaps the bipartite attitude alignment for those with attitude congruence on this item represented participants' movement towards what they perceived to be the most prototypical subsequent attitudes held by people who hold a particular attitude on refugees. If the first attitude question had been different, the form that attitude alignment took may also have been different as the perceived prototypical attitudes of a group revolving around a different attitude may be different. Future research should explore whether attitude alignment with ingroups reflects a convergence towards the attitudes perceived to be most prototypical for that particular attitude-based group. Future research should also explore whether any subsequent attitude expressions are effected by differences in the original attitude around which groups revolve.

We also found that attitude alignment strengthened identification for those interacting with ingroup members. This provides supportive evidence for a dynamic relationship between attitudes and identification, where attitude congruence can foster identification, motivate subsequent attitude

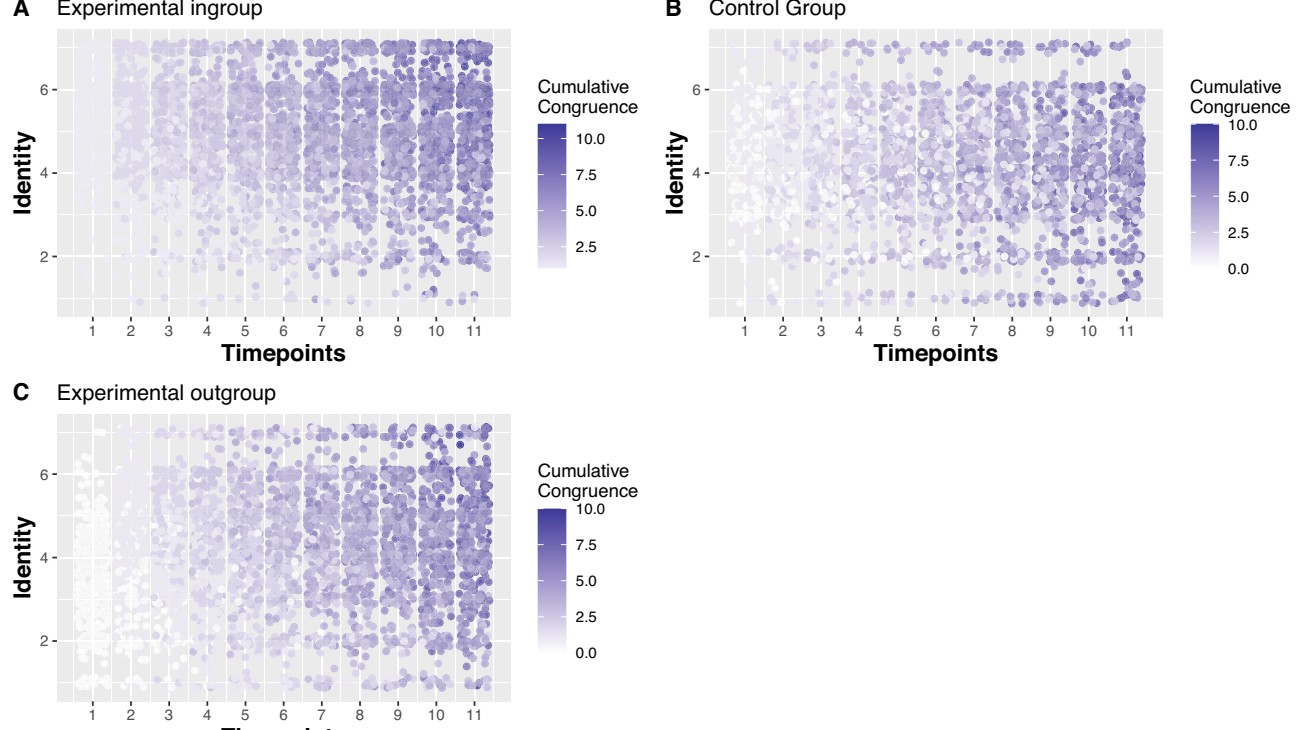

**Fig. 6 | Average identification at each item/time point for each, across condition and congruence levels.** 'Cong' refers to when individuals have congruence with their dyadic partner on a given attitude. 'Incong' refers to when individuals have incongruence with their dyadic partner on a given attitude. Error bars represent 95% confidence intervals (*n* = 836). Green represents the experimental ingroup, yellow represents the experimental outgroup, grey represents the control group. Average identification begins at the highest point in the ingroup and despite differences in congruence over time, congruent outgroup members never reach an average identification as high as any congruent ingroup members. Incongruent ingroup members never reach an average identification level as low as incongruent outgroup members. This displays the impact of the initial opinion-based group identification.

alignment, and where subsequent attitude alignment can strengthen the identification that motivated the alignment in the first place. It also seems that belonging to an opinion-based group provides some resilience to any identity weakening effects of attitude incongruence[10]. Thus, people are active agents in interaction and in the construction of social identities[2–9]. These results provide empirical evidence supporting theoretical models such as the extension of the strategic SIDE[2] Quayle's performative network theory of attitudes[3], Durrheim et al.'s identity performance theory[5] and the social identity approach at large[19].

It seems that participants may have been able to use the knowledge they had of the opinion space to locate their partner in it and performatively [dis] align their own attitudes in a process of joint positioning (although we note that further research is needed here). These results suggest that the attitudes that enter the social world and become normative can be defined in interaction through an active process of attitude co-ordination or polarisation depending on one's interactive audience. Furthermore, identity performance through attitude expression influences the definition and strength of social identity because it constrains which attitudes can be recognised as normative for one's group (the ones that get expressed), and because cumulative attitude alignment strengthens identification[10]. This also implies that attitude alignment/consensus is vulnerable to identity-based polarisation as consensus is identity laden. Thus, performative attitude expression has the potential to change social reality itself[5].

Attitudes are often understood as individual phenomena. Classical 'individualist' perspectives on attitudes (e.g., social cognitive theory) conceptualise attitudes as fixed cognitive structures; as stable, internal ways of thinking about target stimuli relatively unconstrained by social context. In contrast to this classical conceptualisation, our research demonstrates that attitudes are collective constructs jointly produced in interaction and which facilitate the formation, evolution, expression, consolidation, and

mobilisation of shared social identities as they develop in response to contextual factors[2,39]. Our results also support the idea that individual cognition is a function of public attitude expression in line with the original stipulations of the self-categorisation theory. Turner and colleagues state that 'Social reality testing (consensual validation, seeking the agreement of ingroup others) …are interdependent aspects of achieving a valid social cognition. Individual perception and cognition rest on socially validated knowledge, theories, methods, and categories, just as the power of social consensus to define reality for group members makes sense only if the individual views that make up the consensus have been independently tested.'[65] Thus, the dynamic attitude-identity relationship and the identity motivated strategic expression of attitudes should effect individual perceptions and cognition, not merely attitudes expressed publicly.

It is crucial to recognise the dynamic nature of social influence effects, and to recognise that situational identities and expressed attitudes can be actively defined and constructed in interaction[5], are constrained by the ingroup/outgroup relationship between interactants[2], and are important processes that can produce social change[5]. It would be interesting for future research to further explore the idea that people are active agents in the creation of identities which can be fostered in interaction via strategic use of contextually relevant attributes, in this case, attitudes. The experimental method used in study two provides an example of one way in which opinion-identity dynamics in interaction, and identity emergence in the online context, can be explored.

**Practical implications**

Understanding why and how attitudes become aligned is fundamental to understanding processes of social influence, including promoting positive social influence (e.g., vaccine uptake) and resisting negative social influence (misinformation, bot influence). The resolution of many global

problems (such as climate change) requires group attitude alignment to be successful. Speculatively speaking, our results suggest that tackling large scale social issues that require attitude synchronisation for their efficacy (e.g., vaccine uptake) may need to go beyond logical or rational arguments, and consider the group nature of attitudes and attitude interactions. Our results suggest that sharing/expressing commonly held attitudes may heighten the likelihood of attitude alignment on a subsequent target attitude one is hoping to change/influence, as it may foster a sense of ingroupness based on congruence on the first attitude. Furthermore, attitudes shared by others already perceived as ingroup members may be particularly influential.

Of course, the same processes of influence could also be harnessed by those attempting to influence attitudes in a socially harmful way (e.g., vaccine conspiracies) but also to protect against such attempts[66]. These social psychological mechanisms are likely exploited by 'bots' on social media (automated; semi-automated or sock-puppet accounts): a bot that has previously shared attitudes or tweets that we agree with may have particularly strong influence on future attitudes. Our results also suggest that purposeful misinformation can undermine consensus, especially if that misinformation is coming from accounts that have previously shared attitudes congruent with one's own attitudes, or those accounts are perceived to represent ingroup members.

Recognising that people may publicly share attitudes online to produce and consolidate group identity may also help to explain why certain attitudes in particular are spread and go viral. It is crucial to understand how attitude alignment or synchronisation in social networks occurs, as social movements created from such attitude alignment are powerful mobilisers of social change and social action[16]. Important large scale social movements have mobilised on Twitter through attitude expression in the form of hashtags (for example #Metoo; #Notallmen and #BlackLivesMatter; #AllLivesMatter, 17, 16, 48). In study one, we captured a mobilisation of attitudes in response to a situation of threat (the Russian invasion of Ukraine, 2022). The mobilisation we observed on Twitter reflected very strong international support for Ukraine at a particular moment in time. Importantly, the particular attitudes that became aligned and that were spread on social media depended on the network structure, implying that the attitude alignment process can be produced and constructed through interaction and can be constrained by one's audience (those one is connected with on social media). Of course, this research was exploratory and thus future research should verify findings.

In study two, we demonstrated that attitude expression is strategic, and constrained by the identity relationship between interactants. We also found that the opinion space aligned when individuals publicly expressed attitudes in interaction with opinion-based ingroup members. That is, the attitudes that enter the social world, and thus social reality itself, seem to be contingent on the identity relationships between interactants[5]. More generally, the results support the assertion that groupness is central to the attitudinal divides seen in relation to the war in Ukraine. This implies the expression of attitudes via computer-mediated communication is directed by the online behaviour of those we interact with, and identity-laden attitudes should be most amenable to online propagation.

That attitudes are constrained by the group relationship between interactants implies that the attitudes that enter the social world are moulded by the identity performative nature of attitude expression. Thus, the attitudes that gain social power are likely constrained by social identity. This has knock on societal effects as the attitudes that people choose to share on social media have macro-level consequences[48,67,68]. Identity performance increases the likelihood of publicly expressing identity-laden attitudes on social media and in the process changes the agenda for which attitudes have macrolevel outcomes. e.g., sharing attitudes about sanctioning Russia on Twitter likely had knock on effects such as companies pulling out of Russia. Such macrolevel behavioural effects of attitude-based groups are difficult to capture by measuring individual's activism and behavioural intentions. Future research should attempt to assess macrolevel behavioural outcomes using nationally representative longitudinal studies, or using large-scale social media studies tracking attitude expression and its correspondence to the mobilisation of action over time.

The opinion space on social media (and in general life) is constantly being produced and reproduced via interaction. In the current study, we captured attitude alignment and propagation process on an upward trajectory but care needs to be taken to retain such propagation tendencies. With fatigue, motivation for attitude propagation in relation to a particular topic can unwind and other, more topical issues can gain momentum. Similarly, in a different context, or in a different moment in time, the mobilisation process we observed, and the particular attitudes that became consensualised, could have been different. The opinion space can thus reconfigure, resulting in the alignment and propagation of different attitudes. Of course, our results cannot fully explain why particular attitudes gain momentum on social media at certain moments in time, but our results strongly imply that to fully understand such processes we need to account for groupness. Attitudes that clearly reflect or represent group identity may be more amenable to online propagation. Future research should recognise the importance of groupness when attempting to understand virality in the online context.

More speculatively, situations of uncertainty and threat, such as the Russian invasion of Ukraine, may perpetuate attitude propagation. When interacting with an ingroup, we may be motivated towards attitudinal alignment, not only to pursue goals of ingroup inclusion, but also because ingroups have collective power only when they achieve consensus or cohesion on core dimensions. When attitudes become coordinated and collective, they have the power to transform general social opinion on issues (e.g. women's/LQBTQ rights movements) as well as ignite collective action and transform macro level societal structures[4,5]. As group cohesion maximises the collective agency and power of a group[5,12,38,69], the strategic alignment of attitudes should be especially pertinent in situations of threat, such as the Russian invasion of Ukraine. Finding that those who engaged in attitudinal interaction aligned their attitudes more than those who privately expressed their attitudes in the context of the Ukraine crisis, may provide preliminary support that people are motivated to align their attitudes to achieve collective power. Indeed, this would reflect the identity mobilisation dimension of the social identity performance model[2]. Klein and colleagues explain that, not only can identity performance serve to consolidate identity but also to mobilise it. That is, identity performance can serve to foster ingroup related action. It is difficult to ascertain from the current results whether or not people performatively expressed attitudes to mobilise identity, and future research is needed to probe these effects further.

## Limitations
In study one, we captured tweets only in the English language. This is a one-sided, biased representation of online interaction revolving around the war in Ukraine. In particular, it fails to account for the languages of the two countries directly involved in the war. Future research should assess these dynamics accounting for tweets and hashtags that were popular in other languages. Furthermore, we may not have identified all popular English hashtags prior to data collection, thus certain attitudes may have been excluded at the data collection stage. Nonetheless, it is not necessary to capture every attitude expressed online to see a pattern of attitude responses thus, the patterns we observe should not be detrimentally affected by any bias introduced at the data collection stage.

In the current paper, we conceptualise hashtags as attitude expressions. While hashtags can be considered to serve the same communicative function as attitude expressions, it is important to note that this is not always the case (see ref. 70 for a discussion on the communicative functions of hashtags).

Theoretically, we expect that, in line with the social identity performance model[2], people can be motivated to publicly express attitudes online in order to consolidate and mobilise their identity. We expect that people can look to others close to them in a social network and share similar attitudes to them because they are perceived to share an online social identity and because people are motivated to consolidate and mobilise this

identity. It is important to note that, while the online patterns of attitude-sharing we observed in the Twitter study are in line with these theoretical suggestions, we have not ruled out alternative explanations as this was merely an exploratory observational study of large-scale attitude sharing patterns.

In study two we capture dyadic interaction across 11 rounds of attitude sharing. Of course, real-life interaction online often involves more indirect interaction, much larger opinion systems, and with much larger numbers of people. While study two does capture naturally occurring attitudinal interactions it does not fully capture the dynamics of online attitude expression in the real world. Although we do capture these dynamics in the more ecologically valid twitter context, future research should attempt to replicate these results with larger audience sizes in a controlled experimental context.

Furthermore, in real-life contexts where people are repeatedly exposed to streams of opinions (likes, shares, tweets, upvotes etc.), opinion congruence may have a larger impact than our eleven-round experiment implies. This is because frequently repeated small exposures can have large influences in iterative systems via cyclical processes[71]. Future research should assess whether repeated attitude exposure across numerous iterations can compound to have large-scale effects.

Regarding replicability, we expect the process of synchronization of attitudes and identity in groups to be quite universal and highly replicable, although the specific form and functions of synchronization in a given context will be subject to abstract agreements on group norms and self-stereotypes. For example, academics value independence, and therefore disagreement (within bounds, and not on core issues) is normative for such groups in academic contexts. Therefore, we expect the very general process we capture in this paper to be creatively and strategically adapted by different groups in different contexts, but for the process of synchronization on core attitudes to be important for group emergence and identification of individuals to groups in most contexts.

Additionally, attitudes dynamically oscillate in and out of group identities; and opinion-based groups, and inter-group alliances, can be ephemeral[39,65,72]. Therefore, we do not expect this study to replicate directly, at least, not for long. We have captured emergent dynamic opinion-based groups by identifying a topic of great social importance where these processes were clearly at work at the time of the study. When the same design, with the same items, is run at a different moment of history, agreement may not have quite the same import. A full replication would require redesigning the study around whatever issues people are cohering around at that time. On the other hand, we do also expect people to form ephemeral opinion-based groups and identifications on less important, or even arbitrary, topics, as in minimal-group settings[10]. However, people would not be able to recognize allies based on their first answers if they are not already familiar with the emergent group structure related to the topic, and our key manipulation would be less potent.

## Conclusions

These findings provide some insight into how particular attitudes enter the social world, why certain attitudes propagate (go viral), and how social identities are constructed and defined in the online context through attitude expression. Broadly speaking, these results provide evidence to suggest that expressed attitudes are constrained by the group dynamic between attitude expressors and audiences, and that attitude expressors are active agents in the construction of identity, expressing attitudes performatively to construct and consolidate identities. These results also have practical implications, strongly suggesting that online attitude expressions involving attitudes that clearly reflect or represent group identity may be more amenable to online propagation and have particularly strong influence on future attitudes.

## Data availability

All data generated and used in the current studies are available at the following link. (https://osf.io/w8shc/?view_only=34d38b51dc8a4691bf4fec2dbe3cde66).

## Code availability

All analytic code is available at the following link. (https://osf.io/w8shc/?view_only=34d38b51dc8a4691bf4fec2dbe3cde66).

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

## Acknowledgements

We would like to acknowledge and thank Professor Ailish Hannigan for her expertise and advice regarding the statistical methods used in this paper. We also thank the Irish Research Council [grant number GOIPG/2022/545], the Science Foundation Ireland [Grant number 18/CRT/6049], and the European Research Council [grant number 802421, 2020] who provided funding for this work. Note that these funders had no role in study design, data collection and analysis, decision to publish or preparation of the manuscript.

## Author contributions

Caoimhe O'Reilly: Conceptualization, Methodology, Writing – Original draft preparation, Formal analysis, Visualisation, Investigation, Software, Data curation, Project Administration. Shane Mannion: Formal analysis, Visualisation, Data curation, Writing – review and editing Paul Maher: Conceptualization, Methodology, Writing – review and editing, Supervision. Elaine M Smith: Conceptualization, Formal analysis, Writing – review and editing. Pádraig MacCarron: Formal analysis, Visualisation, Writing – review and editing. Michael Quayle: Conceptualization, Methodology, Writing – review and editing, Supervision, Funding acquisition.

## Competing Interests

The authors declare no competing interests.
