## [Peer Review File · Communications Psychology]

6th Jun 23

Dear Ms O'Reilly,

I apologise for the delay in making a decision on your manuscript. As I communicated to you, one of our reviewers has gone unresponsive and we were required to find a replacement - thank you for your patience.

Your manuscript titled "Strategic attitude expressions as identity performance and identity entrepreneurship in interaction" has now been seen by 4 reviewers, and I include their comments at the end of this message. They find your work of interest, but raised some important points. We are interested in the possibility of publishing your study in *Communications Psychology*, but would like to consider your responses to these concerns and assess a revised manuscript before we make a final decision on publication.

We therefore invite you to revise and resubmit your manuscript, along with a point-by-point response to the reviewers. Please highlight all changes in the manuscript text file.

Editorially, we ask you to prioritize the following main issues in revision:

- 1) Reviewers 2, 3, & 4 raise a number of technical concerns. Please provide further analyses to substantiate your interpretation and add further clarifications where necessary.
- 2) Please undertake the necessary textual revisions to address the reviewers' requests for greater clarity in the presentation of the study background, hypotheses, methods, and interpretation.
- 3) Any non-preregistered (exploratory) analysis and deviations from the preregistration need to be clearly labelled as such (all preregistered analyses should be reported).
- 4) As you address the remarks about further necessary analysis and statistics by Reviewers 2 and 4, please ensure you meet our requirements for reporting and interpreting statistics (<https://www.nature.com/commspsychol/submit/submission-guidelines#statistical-guidelines>). All claims about null-effects must be supported with positive evidence for the null, e.g. derived from Bayesian statistics or two-sided tests of equivalence. Please bear in mind these requirements as you revise
- 5) Your manuscript is lacking details on the study's ethics agreement, please ensure transparent reporting in the manuscript and in the Reporting Summary.

Please use the following link to submit your revised manuscript, point-by-point response to the referees' comments (which should be in a separate document to any cover letter) and the completed checklist:

[link redacted]

** This url links to your confidential home page and associated information about manuscripts you

may have submitted or be reviewing for us. If you wish to forward this email to co-authors, please delete the link to your homepage first **

Please do not hesitate to contact me if you have any questions or would like to discuss these revisions further. We look forward to seeing the revised manuscript and thank you for the opportunity to review your work.

Best regards,

Antonia Eisenkoeck

Antonia Eisenkoeck
Senior Editor
Communications Psychology

EDITORIAL POLICIES AND FORMATTING

Editorial Policy: [Policy requirements](https://www.nature.com/documents/nr-editorial-policy-checklist.pdf) (Download the link to your computer as a PDF.)

Furthermore, please align your manuscript with our format requirements, which are summarized on the following checklist:

[Communications Psychology formatting checklist](https://www.nature.com/documents/commspsychol-style-formatting-checklist-article-rr.pdf)

and also in our style and formatting guide [Communications Psychology formatting guide](https://www.nature.com/documents/commspsychol-style-formatting-guide-accept.pdf) .

* TRANSPARENT PEER REVIEW: Communications Psychology uses a transparent peer review system. This means that we publish the editorial decision letters including Reviewers' comments to the authors and the author rebuttal letters online as a supplementary peer review file. However, on author request, confidential information and data can be removed from the published reviewer

reports and rebuttal letters prior to publication. If your manuscript has been previously reviewed at another journal, those Reviewers' comments would not form part of the published peer review file.

* **CODE AVAILABILITY:** All Communications Psychology manuscripts must include a section titled "Code Availability" at the end of the methods section. In the event of publication, we require that the custom analysis code supporting your conclusions is made available in a publicly accessible repository; at publication, we ask you to choose a repository that provides a DOI for the code; the link to the repository and the DOI will need to be included in the Code Availability statement. Publication as Supplementary Information will not suffice. We ask you to prepare code at this stage, to avoid delays later on in the process.

* **DATA AVAILABILITY:**

All Communications Psychology manuscripts must include a section titled "Data Availability" at the end of the Methods section or main text (if no Methods). More information on this policy, is available at <http://www.nature.com/authors/policies/data/data-availability-statements-data-citations.pdf>.

At a minimum the Data availability statement must explain how the data can be obtained and whether there are any restrictions on data sharing. Communications Psychology strongly endorses open sharing of data. If you do make your data openly available, please include in the statement:

We recommend submitting the data to discipline-specific, community-recognized repositories, where possible and a list of recommended repositories is provided at <http://www.nature.com/sdata/policies/repositories>.

If a community resource is unavailable, data can be submitted to generalist repositories such as [figshare](https://figshare.com/) or [Dryad Digital Repository](http://datadryad.org/). Please provide a unique identifier for the data (for example a DOI or a permanent URL) in the data availability statement, if possible. If the repository does not provide identifiers, we encourage authors to supply the search terms that will return the data. For data that have been obtained from publicly available sources, please provide a URL and the specific data product name in the data availability statement. Data with a DOI should be further cited in the methods reference section.

REVIEWERS' EXPERTISE:

Reviewer #1: effect of group on attitudes, social identity

Reviewer #2: effect of group on attitudes, social identity

Reviewer #3: effect of group on attitudes, social identity

Reviewer #4: social network analysis

REVIEWERS' COMMENTS:

Reviewer #2 (Remarks to the Author):

This paper investigates a very interesting question — namely how people align their attitudes with those of ingroup members and what role identity leadership plays in this process. This pertains to a burgeoning area of interest in a range of spheres (not just social and political psychology, but also organizational and health psychology). In line with other recent work in this vein, this paper offers clear insights into these issues that have the capacity to ingratiate and extend a range of previous findings and insights.

Even though the theoretical arguments that inform this research are quite complex, the logic and rationale for the paper was generally very clear, as was the overall case that it presented. Moreover, I think the paper will be accessible to Communications Psychology readers and of considerable interest to them — as well as to a broader readership interested in the issues of leadership, attitudes and collective behaviour that it explores.

It is also worth noting too that while the arguments presented here make very good sense from the perspective of a social identity approach (e.g., Haslam et al., 2011), they are likely to be quite novel to many readers, and accordingly, I think the paper should have considerable value in helping to signal current developments in the field. So while a little bit more work is needed to really underline and sharpen the paper's contribution (see below), I am generally supportive of its publication.

My more substantive comments on the paper are as follows:

1. I thought the paper was generally well written, and that the Introduction was quite easy reading — as indeed was the whole paper. In places, though, the argumentation was a little bit hurried, and I wondered whether readers who were unfamiliar with the social identity approach would be able to 'keep up'. At the same time, there is also a bit of repetition of some points, so the overall point here is that the Introduction could do with a bit of streamlining to make the core narrative easier to follow. More specifically, I think there could be some mileage in (a) being a little bit more forensic in the way that you set the paper up and (b) slowing the exposition down a bit, and taking the reader through the key ideas a bit more slowly. Lines 45-63, for example, contain a number of quite 'big ideas' (covering around three decades of research!) that it would be worth unpacking a bit. I also think that some of the long paragraphs here (and elsewhere) could fruitfully be broken down into smaller chunks. Similarly, I'd note that you don't ever define social identity — or identity entrepreneurship, or the social identity approach — properly so this may leave some readers a bit bemused.

2. This may or may not work, but I wonder what would happen if you tried to represent some of the hypotheses/findings here schematically? An infographic or schematic model might help readers to organize some of your important observations in their heads. I say this because it strikes me that such representations are increasingly used to represent complex results, and also an important way

to communicate research findings (e.g., the “visual abstracts” for papers in *Current Opinion in Psychology*).

3. When it came to the studies I found these pretty interesting at a number of levels (not least the interesting context in which data were gathered). Again, though, the (quite complex) logic of these, and the process through which readers were taken through the results was not as streamlined or as clearly spelled out — and hence as digestible — as it might have been. Certainly, I think more could have been done to flag both the point of the studies, and to talk readers through the theoretical ideas that support them (e.g., explaining what meta-contrast is and how it works). For me, the main issue here was one of story-telling around the separation of wood and trees. At the same time (again), I think the findings here really *were* extremely interesting (e.g., as depicted in Figures 1 and 2), but I fear that as things stand their import may be a bit lost on some readers.

4. The General Discussion was generally very good, and I thought it presented a considered and thoughtful review of the findings, that underlined the key take-away points from the research. In particular, I was persuaded by the general argument, and found nothing here to disagree with (and a lot that was of interest). At the same time though (as per Point 1 above), I thought the general treatment of the findings was quite high-level and that more might have been done to explain to a general audience (comprised of people who are not au fait with the social identity approach) how these findings supported and extended key theoretical ideas, and why this matters at a very basic level. For me, this relates to the capacity for this research to show (in funky ways), how, rather than being fixed cognitive structures (in ways suggested by classical social cognitive theory), attitudes are collective products which help both to build and to express shared social identities as these unfold over time and in response to contextual factors (in ways first argued by Oakes et al., 1994). While this may not be ‘news’ for social identity theorists, I suspect it is for many readers of this journal, and that more might be done to help them embrace both the theory and the metatheory that this research supports.

Minor points

1. I.204 You should say “data were” (data is plural)
2. For Figure 6, I would separate the title of the figure from a note that then talks readers through this.
3. In Table 5 I wouldn’t say “Attitudes asked” but would refer to these as “Stimulus questions”

Reviewer #3 (Remarks to the Author):

Thank you for giving me the opportunity to review the paper “Strategic attitudes as identity performance and identity entrepreneurship in interaction”. There’s much to like about this paper; it is well written and informative and looks at an interesting topic, namely how identity and attitudes co-emerge in interaction.

The work is based on Social Identity Theories, for example, the SIDE model and work by Haslam, Reicher and others.

General comments: The major claim of the paper is that identity and attitudes are dynamically related and are often strategic and motivated to foster ingroup coherence. These claims are interesting and somehow novel (although are also made in work by Klein, Durrheim et al., that are cited heavily in the manuscript). I think that Study 2 provides good and novel insights into the occurrence of attitude coherence and identification processes, but Study 1 doesn't as I will outline below. The work will be an interesting addition to the Social Identity perspective on communication, though it's probably not 'paradigm' shifting or developing a new theory.

Before I can recommend publication, I have a few more specific comments.

Although the introduction/ theoretical background is well written and covers many relevant models/ideas, the deduction of hypothesis seems a bit confusing.

For example, on page 3 the authors write:

The main hypothesis is that Identification can influence future attitude expression as a result of attitude alignment and identity performance.

Then on page 7 they write:

- The main proposal is that the interaction with people perceived as ingroup members will lead to alignment on attitudes.

In my understanding identification and interaction with ingroup members are not identical concepts and it's unclear how these two proposals relate to each other.

Further on page 8 they then write:

Main hypothesis is that greater attitude congruence will be related to greater identification for those expressing attitudes to ingroup members compared to outgroup members.

◇ here identification doesn't feature at all in the hypothesis and it's unclear whether one proposal or several are tested.

Study 1:

I think more care should go into deducting the hypothesis and alignment between ideas. Following from this, it is unclear what Study 1 is showing in relation to the main hypothesis. Neither identification, nor ingroup interaction is clearly 'measured' in this study. In addition, I might need more clarification on the method. First you say you cluster groups/follower networks based on hashtags but then you also use the hashtags to visualise attitude alignment; isn't this slightly tautological as you later also write that you showed that hashtags were different between communities.

You further report that the results showed that individuals tend to publicly express hashtags that others in their community express and that you showed synchronisation. Firstly, my understanding of Twitter is that communication mostly works via Hashtags, thus, expressing hashtags in inherently

part of the behaviour and not surprising; second, I'm not sure how you demonstrate synchronisation, and it would be good to clarify.

More generally, I'm not sure what Study 1 adds and how it relates to either the theorising (as neither identification nor ingroup/outgroup interaction is measured) or the hypothesis.

Study 2:

As Study 2 is quite complex, it might be a possibility to take Study 1 out for more clarification.

On p 20 you write; you excluded participants from Prolific who participated in other studies; are these other studies related to your work here but are not included?

-£1.80 seems not very much for such a long study; how long did participants spend on the study?

Results: I'm not quite sure whether I fully understand how you measured Attitude alignment ((p28) and what the significant different (that you argue shouldn't be significant) means?

Lastly, I'm a bit confused about the concept of identity entrepreneurship in the title. The term was coined by Reicher and Haslam and refers to a leadership process. It might be that it is entailed in dyadic interactions as well but it's not clear how/ where you see it here and how it was conceptualised/operationalised.

Reviewer: Ilka Gleibs

Reviewer #4 (Remarks to the Author):

This paper addresses the concept of social identity performance in two studies. In the first study, the authors show that, in the context of the Russia-Ukraine war, online communities distinct hashtags. The second study, using a quasi-experimental method, examines how attitude alignment as a function of the group membership of the audience & and awareness thereof and how this influences identification. It convincingly shows that such alignment helps consolidate identity in the presence of an ingroup audience.

I very much appreciated reading this paper for a variety of reasons:

- It capitalizes on a major political event to study group behavior "in the wild" but combines this with a much more controlled experimental study.
- This is a strong contribution to one of the dominant approaches to group behavior (the social identity approach) and allows extending it to new perspectives by showing not only how identity expressions are a consequence of the salience of group identities but serve to construct group identities. This idea of "identity construction" is often used in qualitative work and social constructivist perspectives (in sociology, anthropology, etc.). Such has many merits, but often suffers from a difficulty to model psychological processes (with the added risk of subjectivity). Study 2 especially, allows to empirically establish this process of identity construction using a novel and innovative experimental method. This is a major contribution.
- The authors use methods that have been very seldom used in the social identity literature to tackle issues of central interest to them. The use of network analysis to identify communities and the

clustering of attitudes is especially worthy of interest in this regard.

- The findings have wide theoretical and societal implications for how large group of people may come to synchronize attitudes. Specifically, public discourse on mass opinion formation often uses an “epidemiological” metaphor, suggesting that attitudes are spread like viruses without considering how the expression of such attitudes serves distinct needs associated with specific psychological processes. By engaging such processes, the paper offers a fresh and innovative perspective on such mass opinion formation.

- I appreciated that study 2 was preregistered and that the authors made their material and analyses transparent.

This notwithstanding, I have a few comments that may be worth addressing to make the paper stronger. I am sharing them below in no particular order of importance.

While I like both studies, it is important to note that they tackle quite different research questions. In this regard, playing the devil’s advocate, I wondered whether the results of study 1 could be interpreted as reflecting some form of “ordinary” conformism with groups? The difference would be that people just conform to the norm consisting in the expression of attitudes by other ingroup members (without considering how this affects their audience) whereas the SIP framework accords a major role to the audience. It would be useful to clearly articulate why the SIP framework offers a better account of these trends a simple conformism account.

The results of study 1 could be interpreted either in “inductive” (i.e., communities form bottom-up based on the expression of common views – through an aggregation mechanism – homophily driving group formation) or “deductive” (i.e., people express views that are consistent with those expressed by other members of the communities because they identify with these – this would be bottom-up). The paper seems to make a case for the latter view, which is consistent with self-categorization theory (and the social identity performance approach). But it is not clear to me that the proposed analyses allow to clearly disentangle these two possible explanations. I assume that by doing analyses over time (i.e., showing that the communities pre-exist the use of the tags), it would be possible to more convincingly do so. It may also be possible to respond to the above question.

I am not an expert in network analysis. Upon reading the results, I found that I needed some clarification on several aspects. Table 1 and 2 only report crosses to describe the association between hashtags and communities. Why not report actual frequencies? This would allow computing inferential statistics and also apply descriptive methods allowing to map the communities on the hashtags (thinking of factorial correspondence analysis as a possibility). It may also be useful to explain in more detail the meaning of the network indicators used in study 2 (i.e., clustering, path length, edges) and how they adequately operationalize the constructs of interest. I had to trust the authors’ knowledge of the subject to be convinced here (to avoid cluttering the MS, a more extensive explanation could be provided in the SM or a source easily understandable by non-experts pointed to).

With respect to the analysis of study 2, the authors relied on mixed models. More information should be provided on how these models were specified and why (i.e., in terms of random factors especially) without having to read the syntax.

Something that may be worth clarifying regarding study 2 pertains to the role of the quasi-experimental factor. Given that participants were not randomly allocated to ingroup and outgroup

(which makes perfect sense here), is it possible that a confounded factor, such as the content of the attitude play a role in some of the results? Thus, if Prolific participants tend to be “pro-Ukraine”, it is likely that pro-Ukraine attitudes are more represented in the experimental ingroup than in the experimental out-group condition (it would be less likely to find two pro-Russia participants in the same dyad than two pro-Ukraine). In this regard, it would be useful to report the “raw” attitudes (I assume the items form a scale) in each group. Further, if this is the case, I wonder to what extent the clustering observed in the ingroup audience condition may be in part due to the ingroup condition being composed of a more homogeneous group in the first place, reflecting a form of polarization. This may be worth discussing.

I found it particularly interesting, and consistent with SCT, that the outgroup audience leads to attitude misalignment. I am puzzled however by the finding that such misalignment is not associated with greater identification. Using a SCT framework, one would probably expect that expressing views that differentiate the ingroup from the outgroup (in a very salient intergroup context) should lead to higher identification. This should be discussed.

In this regard, a possibility that should be considered is that, when responding to the identification scale, people are still “performing” their identity rather than expressing a “private” view. If this is the case, downplaying one’s identity in front of a person discovered to hold a very different attitude from one’s own (outgroup) may be a strategy to facilitate interpersonal interactions. In this respect, it would be useful to report whether participants were sure that their responses were not visible to the audience?

In our paper (Klein et al., 2007), that the authors use as one of the main foundations of this work, we highlight two possible roles of social identity performance: one is consolidation, the other mobilization. The authors focus on the first function. However, they show that the outcome of this function is actually successful mobilization. It would be interesting in the discussion to discuss this possible paradox and/or question the value of this distinction.

My name (“Klein”) is misspelled on several occasions as Klien.

Figure 3 comes straight from SPSS and could be improved visually.

Something that I found puzzling is that so many countries are represented in the sample although the native language of the participants was English. Is this not an error?

I very much hope these comments will be helpful in the perspective of a possible revision.

Reviewer #5 (Remarks to the Author):

I have now had the chance to read and review “Strategic attitude expressions as identity performance an identity entrepreneurship in interaction”. I was asked to review the social network analysis (SNA) portion of this manuscript. This is a good thing because I only have a cursory knowledge about theories of social identity.

#1 Network boundary

Perhaps the first step in collecting network data is to determine the boundary. That is, who is included and not included in the network. The inclusion criteria in the current manuscript is “English containing hashtags relating to the Russia/Ukraine war, starting from the beginning of 2022 the invasion (24/02/22) and ending the day we commenced gathering data (28/06/22)”. A couple of questions.

What was the full list of hashtags? I tried to open the Supplementary Materials document on the OSF page, but it is just a blank document. Perhaps this is just a problem with OSF?

Second, what is the rationale for including the specific hashtags? What was the process of determining which to include and which not to include? For instance, many of the hashtags express a valence on the war (e.g., #stoprussia), some are incredibly general (e.g., #peace), and some strike me as irrelevant (e.g., #ethiopia—am I missing something here?).

Finally, isn't a bit surprising that the data only captured 8,149 users? This was one of the biggest stories of 2022. It strikes a red flag regarding validity if the inclusion criteria only was able to extract that amount of users. Perhaps many people tweeting about the war just didn't use hashtags?

#2 No rationale for undirected network

Twitter is an undirected network. That is, just because you follow me, doesn't mean I follow you back (unlike Facebook friends). As such, I see no point in analyzing the network as undirected because we are then artificially creating networks. This especially true because further analysis is done on the network as directed. I highly advocate removing this analysis because it could provide misleading interpretations of the data.

#3 Why largest component

Unless I am mistaken, there is no assumption in the Infomap community detection algorithm that the network be a single component. As such, what is the rationale for only applying the algorithm to largest connected component? Are we not losing information on the more peripheral regions of the network? This unanalyzed region seems to be pretty big too if only 3812 nodes were analyzed.

#4 Footnote 2

Four hashtags were removed because they were “not that informative”. What makes a hashtag informative? What is the criteria? Why are the ones included in the tables informative and others not?

#5 Lack of statistical evidence

The manuscript writes that “These results demonstrate that individuals tend to publicly express hashtags that others in their follower communities also express.” However, what information from Table 2 gives the reader evidence that this is a correct interpretation? For instance, if the null hypothesis was true (communities don't use the same hashtags), what would the table look like?

This data seems ripe for some sort of statistical analysis. For instance, in one column we have a categorical variable (community membership) and the other is a multivariate continuous variable (number of times a user used a particular hashtag). Essentially, we want to see if hashtag use has an association with community membership. Perhaps analyzing on the individual level makes sense. That is, use community membership as a categorical variable to predict each hashtag use as a continuous variable. An ANOVA for each hashtag should do the trick, no?

23 June 2023

Re: Submission of Revision Communications Psychology “Strategic attitude expressions as identity performance and identity creation in interaction”

Dear reviewers,

We appreciated your insightful and constructive input the on the previous submission of this manuscript. We have addressed all of the concerns highlighted in a substantive revision and detail how they are addressed below. We provide verbatim text from the revised manuscript which is distinguished by red font. We now submit what we feel is a much improved manuscript.

Sincerely,

The authors

6 R2	I thought the paper was generally well written, and that the Introduction was quite easy reading – as indeed was the whole paper. In places, though, the argumentation was a little bit hurried, and I wondered whether readers who were unfamiliar with the social identity approach would be able to ‘keep up’. At the same time, there is also a bit of repetition of some points, so the overall point here is that the Introduction could do with a bit of streamlining to make the core narrative easier to follow. More specifically, I think there could be some mileage in (a) being a little bit more forensic in the way that you set the paper up and (b) slowing the exposition down a bit, and taking the reader through the key ideas a bit more slowly. Lines 45-63, for example, contain a number of quite ‘big ideas’ (covering around three decades of research!) that it would be worth unpacking a bit. I also think that some of the long paragraphs here (and elsewhere) could fruitfully be broken down into smaller chunks. Similarly, I’d note that you don’t ever define social identity — or identity entrepreneurship, or the social identity approach — properly so this may leave some readers a bit bemused.	We have added a new section at the beginning of the introduction to introduce the core theories and concepts from the outset to ensure that those who were not familiar with the social identity approach could follow our rationale. In particular we outline the social identity approach, define social identity, and outline the transition from Turner’s original perspective to the acknowledgement by social identity theorists of the performative and active nature of social identities (Reicher et al., 2012). We have also gone through the document and broken any large paragraphs into smaller chunks. Page 4 “The Social Identity Approach In order to understand the theoretical rationale for the proposals in the current paper, we first must outline the theoretical model on which the rationale for these proposals is based: social identity approach. The social identity approach (19, 20) asserts that the implicit or explicit presence of others in a particular comparative context can imbue categories with social meaning explaining that “the mere perception of common category membership” is sufficient and necessary for social identification and resultant group formation (21, p. 3). That is, beyond the effect of mere similarity (22), the intergroup context and awareness of a category one belongs to and another that one does not belong to, gives one’s own category social meaning and thus it moves from being a mere category to being a social identity. Social identity is defined by Tajfel as “the individual’s knowledge that he [or she] belongs to certain social groups together with some emotional and value significance to him [or
-----------------	---	---

		her] of this group membership” (23 p. 21). Social identification was defined by Turner as “any social categorisation used by a person to define him-or herself and others” or “ the process whereby an individual internalises some form of social categorisation so that it becomes a component of the self concept, whether long-lasting or ephemeral” (20). The self categorisation theory explains that the process via which social categories become internalised as social identities is context dependent and varies (partly) as a function of the meta-contrast principle. The metacontrast principle stipulates that people cognitively structure social categories by adhering to elements in a given context that maximise relative differences between one’s ingroup and one’s outgroup and maximise the relative entitativity of one’s ingroup (20). In the late 90’s and early 2000’s there was a shift in social identity theorising towards recognising the performative, active nature of categorisation and social identity processes (24; 6, 8, 9; 25). The notion of the purposeful deployment of categorisation was introduced, and theorists proposed that categories could be actively created, defined, and deployed (2; 24; 6; 8; 9; 7; 26). That is, theorists began to recognise that categorisation is not something that just passively happens to people, but people have agency, ability, and motivation to be active agents in the construction of categories around which identities and groups revolve. Not only do “social identities reflect social reality” but also social identities move people to create social reality” (9, p. 365). The social identity model of deindividuation effects (SIDE model) recognises the strategic nature of social identity (2; 26) providing a theoretical framework for understanding how social identities can influence and create social reality via motivated identity performance.”
7 R2	2.This may or may not work, but I wonder what would happen if you tried to represent some of the hypotheses/findings here schematically? An infographic or schematic model might help readers to organize some of your important observations in their heads. I say this because it strikes me that such representations are increasingly used to represent complex results, and also an important way to communicate research findings (e.g., the “visual abstracts” for papers in Current Opinion in Psychology).	We have now represented the main hypothesis and results schematically. We note that this journal does not allow graphical abstracts (not sure if this extends to schematic representations). We will therefore leave it to the editor to decide whether we should include these or not.

Attitude congruence fosters group identification

In group interaction motivates attitude alignment
Outgroup interaction motivates attitude disalignment

8
R2

3. When it came to the studies I found these pretty interesting at a number of levels (not least the interesting context in which data were gathered). Again, though, the (quite complex) logic of these, and the process through which readers were taken through the results was not as streamlined or as clearly spelled out — and hence as digestible — as it might have been. Certainly, I think more could have been done to flag both the point of the studies, and to talk readers through the theoretical ideas that support them (e.g., explaining what meta-contrast is and how it works). For me, the main issue here was one of story-telling around the separation of wood and trees. At the same time (again), I think the findings here really *were* extremely interesting (e.g., as depicted in Figures 1 and 2), but I fear that as things stand their import may be a bit lost on some readers.

We have updated the introduction section to more clearly outline the theoretical rationale behind study one. We have updated the mini introduction that specifically relates to study one. We have updated the results and discussion section to more clearly explain how the theory relates to the findings of study one. We have properly explained the metacontrast theory in the introduction. We added a new section to clarify the theoretical ideas supporting study 2 in the mini introduction on page 19

Introduction Page 8

“Performative Attitude Expression in the Online Context

Qualitative research has found that social identity can be actively [re]constructed and [re]defined on social media and that social media can be used as a tool to display and express one’s social identity (Karizat et al., 2021; Jenkins et al., 2016; Yau et

al., 2019; Talbot et al., 2022). Furthermore, previous research has found that, when expressing attitudes online, people are aware of and attentive to their audience (Marwick & boyd, 2010; boyd, 2006; Ellison et al., 2006). That is, the influence of the audience on communication translates to the online context (Mishra et al., 2017). Although people can comprehend the limitless nature of the social media audience, people nonetheless tend to behave as if the audience is more confined and imagine a particular audience when posting on social media (Marwick & Boyd, 2010; Litt & Hargitai, 2016).

In line with the social identity approach (19, 20), in the online context (as in the offline context) people should be motivated to be similar to their online ingroups (20). Posting an attitude online is a social act that goes beyond any similarity motivations however. That is, it is possible to be referentially influenced by one's online ingroups without needing to post anything online (20). The social act of sharing one's attitude online goes further than mere referent informational influence (20). In line with the social identity performance model (2) we propose that people are motivated to publicly express attitudes online in order to consolidate and mobilise their identity (2; 60).

People may look to others close to them in a social network and share similar attitudes to them because they are perceived to share an online social identity and because people are motivated to consolidate and mobilise this identity.

In line with our main proposal (that interaction with people perceived as ingroup members will lead to alignment on attitudes) we also explore whether online communities tend to express similar attitudes as those close to them in the social network. The purpose of this exploration is to determine whether patterns of attitudinal expression in the large scale online social media context suggest that people could be performatively express attitudes similar to their online groups to consolidate their online identities. We do not directly test the motivations behind these patterns, we simply explore and observe large-scale patterns of attitude expression within online communities on Twitter. “

Introduction metacontrast theory explanation Page 4

“The self categorisation theory explains that the process via which social categories become internalised as social identities is context dependent and varies (partly) as a function of the meta-contrast principle. The metacontrast

principle stipulates that people cognitively structure social categories by adhering to elements in a given context that maximise relative differences between one's ingroup and one's outgroup and maximise the relative entitativity of one's ingroup. “

Page 5

“Self-categorisation theory asserts that, in an intergroup context, in line with the metacontrast principle, people should be motivated to be more similar to their ingroup to maximise the relative differences between themselves and their outgroup and to maximise the entitativity of their ingroup (20)”

Study one mini introduction Page 11

“As outlined in the introduction, the social identity performance model asserts that people are motivated to consolidate their identity publicly. We suggest that one way people can achieve this is by publicly expressing attitudes that align with the attitudes of our ingroup members. Public attitude expression is ubiquitous in the online context. The main purpose of study one is to observe large-scale patterns of attitude sharing in ‘real world’ online contexts (as opposed to the more constrained experimental settings) and to explore how patterns of attitudinal expression might relate to the attitudes expressed by others close to oneself in online followership networks. In exploratory analysis, we assess whether individuals tend to publicly share attitudes (in the form of tweets) that are similar to others in their followership networks on Twitter. To do this, we tracked online expressions of attitudes in the form of hashtags on Twitter that related to the Russo-Ukraine war for the five months following Russia's invasion of Ukraine (February, 2022).”

Results and discussion study one Page 14

“This implies that public expression of attitudes on twitter is affected by one's follower network - as different clusters of followers publicly expressed tweets that were qualitatively similar to others close to them in the follower network. According to the social identity performance model and the social identity approach at large, and the social identity performance model, people should be motivated to share attitudes that are similar to their ingroups in order to consolidate and mobilise their identity. The attitude expression patterns that we have observed in the Twitter context in the five month aftermath of Russia's first invasion of Ukraine are consistent with this interpretation as people tended to share

		attitudes similar to those closest to them in the twitter followership network, Thus, using large-scale, ‘real world’, naturalistic data we have observed patterns of attitude sharing consistent with the social identity performance model implying that people’s public attitude expression is performative and constrained by identity motivations. Nonetheless, we have not directly measured or manipulated identification thus we can only speculate why these attitudinal expression patterns have occurred.” Mini introduction study 2 page 19 “‘As outlined in the introduction, in (implicit or explicit) intergroup contexts, people are motivated to be similar to their ingroups to maximise intergroup differences and intragroup entitativity (metacontrast theory; 20), and because ingroups are perceived to hold appropriate norms and accurate information (referent informational influence theory; 20). Furthermore, people are motivated to overtly display their identities via identity performance to consolidate and mobilise these identities (social identity performance model; 2). In study two, we therefore propose that ingroup members should be motivated towards attitude alignment in line with the metacontrast theory, the referent informational influence theory, and the social identity performance model. Our main preregistered hypothesis (hypothesis one) predicted that attitude alignment would be greater in the ingroup experimental condition compared to the control condition and the outgroup experimental condition. Our secondary hypotheses predicted that; hypothesis two: condition (ingroup vs outgroup vs control) would moderate the effect of congruence on identification. We expected congruence to predict identification better in the experimental (ingroup) condition, versus the outgroup and control, conditions. Hypothesis three predicted that change in congruence would predict change in identification over time (greater congruence would be related to greater identification) for the ingroup experimental condition. We expected congruence to predict identification better in the ingroup experimental, versus the outgroup and control conditions. ”
9 R2	4. The General Discussion was generally very good, and I thought it presented a considered and thoughtful review of the findings, that underlined the key take-away points from the research. In particular, I was persuaded by the general argument, and found nothing here to disagree with (and a lot that was of interest). At the same time though (as per Point 1 above), I thought the general	We have made several additions to the general discussion as suggested by reviewer 2. Page 45

treatment of the findings was quite high-level and that more might have been done to explain to a general audience (comprised of people who are not au fait with the social identity approach) how these findings supported and extended key theoretical ideas, and why this matters at a very basic level. For me, this relates to the capacity for this research to show (in funky ways), how, rather than being fixed cognitive structures (in ways suggested by classical social cognitive theory), attitudes are collective products which help both to build and to express shared social identities as these unfold over time and in response to contextual factors (in ways first argued by Oakes et al., 1994). While this may not be 'news' for social identity theorists, I suspect it is for many readers of this journal, and that more might be done to help them embrace both the theory and the metatheory that this research supports.

“Attitudes are often understood as individual phenomena. Classical “individualist” perspectives on attitudes (e.g., social cognitive theory) conceptualise attitudes as fixed cognitive structures; as stable, internal ways of thinking about target stimuli relatively unconstrained by social context. In contrast to this classical conceptualisation, our research demonstrates that attitudes are collective constructs jointly produced in interaction and which facilitate the formation, evolution, expression consolidation, and mobilisation of shared social identities as they develop in response to contextual factors (2; Oakes et al., 1994).

It also supports the idea that individual cognition is a function of public attitude expression in line with the original stipulations of the SCT. Turner and colleagues state that “Social reality testing (consensual validation, seeking the agreement of ingroup others) ...are interdependent aspects of achieving a valid social cognition. Individual perception and cognition rest on socially validated knowledge, theories, methods, and categories, just as the power of social consensus to define reality for group members makes sense only if the individual views that make up the consensus have been independently tested.” (Turner et al., 1994, p.). Thus, the dynamic attitude-identity relationship and the identity motivated strategic expression of attitudes should effect individual perceptions and cognition, not merely attitudes expressed publicly”

Page 44

Bipartite alignment

“That the entire ingroup aligned their attitudes after interaction with only one ingroup member, could potentially reflect individual ingroup members’ strategic tailoring of expressed attitudes to be consistent with the attitude they perceive to be most prototypical. That is, bipartite attitude alignment may be representing a shift towards perceived ingroup prototypical attitudes. In our study, people began with either congruence or incongruence on their attitude about whether their country of origin should extend unlimited welcome to refugees. Perhaps the bipartite attitude alignment for those with attitude congruence on this item represented participants’ movement towards what they perceived to be the most prototypical subsequent attitudes held by people who hold a particular attitude on refugees. If the first attitude question had been different, the form

		that attitude alignment took may also have been different as the perceived prototypical attitudes of a group revolving around a different attitude may be different. Future research should explore whether attitude alignment with ingroups reflects a convergence towards the attitudes perceived to be most prototypical for that particular attitude-based group. Future research should also explore whether any subsequent attitude expressions are effected by differences in the original attitude around which groups revolve.” Page 48 Macrolevel agenda “Attitude based identity performance likely has macrolevel consequences. That attitudes are constrained by the group relationship between interactants implies that the attitudes that enter the social world are moulded by the identity performative nature of attitude expression. Thus, the attitudes that gain social power are likely constrained by social identity. This has knock on societal effects as the attitudes that people choose to share on social media have macro-level consequences (Hopp et al., 2021; Morgan et al., 2010; Brady et al., 2020; Moojiman et al., 2018). Identity performance increases the likelihood of publicly expressing identity laden attitudes on social media and in the process change the agenda for which attitudes have macrolevel outcomes. e.g., sharing attitudes about sanctioning Russia on Twitter likely had knock on effects such as companies pulling out of Russia. Behavioural effects of attitude based groups that are difficult to capture by measuring individual’s activism and behavioural intentions. Future research should attempt to assess macrolevel behavioural outcomes using nationally representative longitudinal studies, or using large scale social media studies tracking attitude expression and its correspondence to the mobilisation of action over time.”
10 R3	Although the introduction/ theoretical background is well written and covers many relevant models/ideas, the deduction of hypothesis seems a bit confusing. For example, on page 3 the authors write: The main hypothesis is that Identification can influence future attitude expression as a result of attitude alignment and identity performance. Then on page 7 they write: - The main proposal is that the interaction with people perceived as ingroup members will lead to alignment on attitudes.	It is true that the statements of the main hypothesis on page 3 and page 7 were slightly different, thanks to the reviewer for spotting this. We have amended the statement on page 3 to be in line with the statement on page 7 and the preregistered hypothesis. Pg 3 “The main hypothesis is that interaction with such ingroup members can influence future attitude expression as a result of attitude alignment and identity performance.”

	In my understanding identification and interaction with ingroup members are not identical concepts and it's unclear how these two proposals relate to each other. Further on page 8 they then write: Main hypothesis is that greater attitude congruence will be related to greater identification for those expressing attitudes to ingroup members compared to outgroup members. à here identification doesn't feature at all in the hypothesis and it's unclear whether one proposal or several are tested.	The reviewer points out that identification and interaction with ingroup members are not identical concepts. We absolutely agree (hence we measure identification and change in identification over time in our studies). This was an error of phrasing rather than a theoretical conflation which we have now amended. On page 8 we are referring to hypothesis two (not the main hypothesis). We do not mention 'main hypothesis' here, the direct quote from page 8 is as follows "We hypothesise that greater congruence will be related to greater identification for those expressing attitudes to ingroup members compared to outgroup members and those privately expressing attitudes." For this hypothesis, total identification (averaged across all time points) is the dependent variable. Here we test one main proposal which is an interaction effect (attitude congruence will be positively related to identification for the experimental ingroup condition more so than the experimental outgroup condition).
11 R3	Study 1: I might need more clarification on the method. First you say you cluster groups/follower networks based on hashtags but then you also use the hashtags to visualise attitude alignment; isn't this slightly tautological as you later also write that you showed that hashtags were different between communities. You further report that the results showed that individuals tend to publicly express hastags that others in their community express and that you showed synchronisation. Firstly, my understanding of Twitter is that communication mostly works via Hastags, thus, expressing hashtags in inherently part of the behaviour and not surprising; second, I'm not sure how you demonstrate synchronisation, and it would be good to clarify.	Many thanks to the reviewer for this comment. We did not outline our procedure clearly enough which has led to some confusion. We have hopefully resolved this. The inclusion criteria for being in the sample was that users had tweeted hashtags related to the Ukraine war. We then generated a follower network which was formed based on followership links between these users. Thus, hashtags are independent of the network visualisation and the community detection is purely based on the network followership structure. It is the community detection (of followership links) that we visualise in Figure 1. We then look at the most popular five hashtags within each of these follwership communities which is represented visually and descriptively in table 1 and supplementary table 1 and find that the most popular hashtags used by each followership community are different. Page 12 “Participants and Procedure We searched twitter to identify popular hashtags related to the ongoing invasion of the Ukraine, as well as hashtags that were ‘trending’

at the time (see Supplementary Note 2 for full list of hashtags), attempting to identify as many dimensions of the online discourse as possible. Participants were included in the dataset if they had tweeted any of the hashtags on the hashtag list between February and June 2022 (from Russia's invasion of Ukraine and for five months after this). This yielded 5178508 tweets from 1053860 users. We then examined the frequencies of hashtag expressions for each tweet and chose six hashtags that had high tweet or retweet frequencies. Our final sample consisted of a subset of users who had tweeted one of these hashtags (Supplementary Note 4). We then created a dataset of all tweets in the original dataset from users who were in our final sample. Next, we gathered the follower list for each of these users, discarding the data for any user whose account information was unavailable through the Twitter API (e.g., restricted, banned, or deleted accounts). We also discarded any user in the follower lists for whom we had no tweets. This left us with 8,149 users. We then built a followership network of these users, where users were linked to every user they were followed by, if that user had also tweeted or retweeted one of the hashtags.

The followership network of Twitter users is a directed network (i.e., there is a directionality associated with followership, unlike, for example, Facebook friendships). We thus employed the Infomap community detection algorithm to the directed version of the network (39). The Infomap algorithm identifies communities based on how information may flow through the network, taking into account the directionality of the followership edges (41). We then identified the five most common hashtags shared in each community detected by both algorithms.”

What is surprising about hashtag expression is that people tend to use hashtags similar to others close to them in their social network i.e., in the Twitter followership network. See below for further analysis that we have conducted which help to better demonstrate this synchronisation of hashtag use within communities.

Page 15

“We then compared the distribution of each hashtag across communities statistically, and in each case a significant chi square goodness of fit statistic demonstrated that the distribution of hashtags was not even across communities and hashtag use characterises these communities

		(Supplementary Analysis 1). This demonstrated that the results depicted in Table 1 differ significantly from a random distribution. We also ran individual ANOVA's to assess the frequency of individuals' hashtag use across all communities. 20 out of 21 showed statistically significant differences (Supplementary Analyses 2). Thus, followership communities on twitter are characterized and clearly distinguished by the combination of hashtags that they publicly express. Taken together, these results all suggest that groups of followers tend to use similar hashtags depending on what "group/community" they are part of. "
12 R3	It is unclear what Study 1 is showing in relation to the main hypothesis. Neither identification, nor ingroup interaction is clearly 'measured' in this study More generally, I'm not sure what Study 1 adds and how it relates to either the theorising (as neither identification nor ingroup/outgroup interaction is measured) or the hypothesis. As Study 2 is quite complex, it might be a possibility to take Study 1 out for more clarification.	Thanks to the reviewer for this comment. Study one demonstrates that people tend to publicly express attitudes that are similar to those closest to them in a followership twitter network. Study one uses naturalistic twitter data which captures genuine, 'real life' attitude sharing (as opposed to the more contrived experimental setting). This brings with it both advantages and disadvantages. Using twitter data means that we are unable to measure identification or ingroup/outgroup interaction. This data is valuable for other reasons, however. What we can do is observe patterns of 'real world', naturalistic, large scale attitude sharing and use our knowledge of social psychology to hypothesise why such patterns have occurred. We can then test such hypotheses in more controlled, stringent experimental environments allowing us to measure the variables we theorise as having some effect on the processes we observe (as we did in study 2). Thus, study one allows us to observe that people tend to share similar attitudes to those close to them in a followership network. Although we cannot test whether this is a direct result of ingroup membership and social identification, what we can say is that people tend to share attitudes similar to those close to oneself in 'real life' social networks – something that we cannot easily manipulate in the lab setting. Thus, study one compliments study two by demonstrating a pattern of behaviour consistent with study two findings in a naturalistic, large scale, 'real life' setting.

Based on the comments from the other reviewers, especially reviewers two and four who particularly seem to appreciate the contribution of study one we would rather retain this study. Importantly, while reviewer four likes the study he also had some requests for clarification (see comment 20).

Please also note that study one was exploratory so we did not have a hypothesis. Nonetheless we should have been clearer regarding the purpose of this study and have therefore added a new section in the introduction

Page 8

“Qualitative research has found that social identity can be actively [re]constructed and [re]defined on social media and that social media can be used as a tool to display and express one’s social identity (Karizat et al., 2021; Jenkins et al., 2016; Yau et al., 2019; Talbot et al., 2022). Furthermore, previous research has found that, when expressing attitudes online, people are aware of and attentive to their audience (Marwick & boyd, 2010; boyd, 2006; Ellison et al., 2006). That is, the influence of the audience on communication translates to the online context (Mishra et al., 2017). Although people can comprehend the limitless nature of the social media audience, people nonetheless tend to behave as if the audience is more confined and imagine a particular audience when posting on social media (Marwick & Boyd, 2010; Litt & Hargitai, 2016). In line with the social identity approach (19, 20), in the online context (as in the offline context) people should be motivated to be similar to their online ingroups (20). Posting an attitude online is a social act that goes beyond any similarity motivations however. That is, it is possible to be referentially influenced by one’s online ingroups without needing to post anything online (20). The social act of sharing one’s attitude online goes further than mere referent informational influence . In line with the social identity performance model (2) we propose that people are motivated to publicly express attitudes online in order to consolidate and

		mobilise their identity (2; 60). People may look to others close to them in a social network and share similar attitudes to them because they are perceived to share an online social identity and because people are motivated to consolidate and mobilise this identity. In line with our main proposal (that interaction with people perceived as ingroup members will lead to alignment on attitudes) we also explore whether online communities tend to express similar attitudes as those close to them in the social network. The purpose of this exploration is to determine whether patterns of attitudinal expression in the large scale online social media context suggest that people could be performatively express attitudes similar to their online groups to consolidate their online identities. We do not directly test the motivations behind these patterns , we simply explore and observe large-scale patterns of attitude expression within online communities on Twitter.”
13 R3	On p 20 you write; you excluded participants from Prolific who participated in other studies; are these other studies related to your work here but are not included?	We excluded people on prolific who had participated in any of our previous studies. These studies are not related to the current work, but rather, relate to previously published work we have done on the attitude/identity relationship.
14 R3	£1.80 seems not very much for such a long study; how long did participants spend on the study?	This was the price suggested by the prolific website as being a fair price for the median time taken to complete the study which was 17 minutes.
15 R3	Results: I’m not quite sure whether I fully understand how you measured Attitude alignment ((p28) and what the significant different (that you argue shouldn’t be significant) means?	We assume here that the reviewer means ‘bipartite attitude alignment’ as this is the section on page 18. Importantly, we have corrected a typo on page 18 which may have led to this confusion. Page 32 “Results demonstrated that the experimental outgroup has statistically significantly shorter path lengths ($M = 4.63$) than the experimental ingroup ($M = 4.67$), $p < .001$).” In acknowledgement that our original explanations may not have been clear enough we have also adapted the original text. Page 26 “As well as assessing attitude alignment within dyads, we also assessed attitude alignment within conditions. To do this, we created bipartite graphs which connected users via attitude agreement (54,

55). In these bipartite graphs there were two types of nodes, one representing attitudes, and the other representing each participant. Participant nodes were linked to attitude nodes via positive edges if a participant agreed with an attitude, or via negative edges if a participant disagreed with an attitude. Next, following MacCarron et al., 2020 (54), network projections were created which connected each participant to other participants based on how much attitude congruence they had with each other (54; 21). Participants would be very close in the network if they had lots of the same attitudes, and further away if they had lots of different attitudes. We created four network projections (bipartite graphs), one for each condition. We then compared the attitude alignment across conditions statistically.”

Page 31

“As well as assessing attitude alignment within dyads, we also assessed overall attitude alignment within each condition via generating bipartite graphs which connect participants via the attitudes they share (Figure 3) (54, 55). Firstly, we connected participants who had congruence on all 11 items. We then lowered the agreement threshold until a majority of participants were in a giant connected component. The threshold was nine items. We then assessed two network quantities; path lengths and clustering coefficients. The path length assesses how "coherent" the attitude space is, so shorter path lengths (shorter distances between nodes) represent higher attitude congruence, and longer path lengths (longer distances between nodes) represent lower attitude congruence. We then assessed whether the average path lengths differed across condition via one-way ANOVA.

Results demonstrated that the experimental outgroup has statistically significantly shorter path lengths ($M = 4.63$) than the experimental ingroup ($M = 4.67$), $p < .001$). This does not support our preregistered hypothesis, however, path length analysis is sensitive to unequal sample sizes and there were 502 people in the experimental ingroup and 434 in the experimental outgroup. The longest path length possible was 15 and the shortest was 1. Such a small difference between means (.03), although statistically significant, because of the unequal sample sizes, is likely to reflect the path length's sensitivity to sample size rather than reflecting any meaningful difference. Clearly, more work is needed to develop network methods to assess psychologically relevant features, and

		real life data which is inevitably susceptible to unequal sample sizes.” Bipartite attitude alignment was a measure of attitude similarity between participants of each condition. That is, we assessed how much attitude alignment there was in all four conditions and compared them to see which condition had the highest overall alignment. To do this we created bipartite graphs (see MacCarron et al., 2020). This graph type contains two types of nodes, one representing attitudes and the other representing people. These two types of nodes get linked with positive edges if a person agrees with an attitude and negative edges if a person disagrees with an attitude. Next, network projections were created which connected each participant to other participants based on how much attitude congruence they had. Participants would be very close in the network if they had lots of the same attitudes, and further away if they had lots of different attitudes. We created four network projections, one for each condition. We then compared the attitude alignment across conditions statistically by comparing two network quantities; path lengths and clustering coefficients. One way of assessing attitude alignment differences between the network projections is to assess path length differences. Shorter path lengths (shorter distances between nodes) reflects higher attitude alignment whereas longer path lengths reflects lower attitude alignment. This type of analysis is very sensitive to unequal sample sizes however, and we had unequal sample sizes between the experimental and control conditions. Using an ANOVA, we did find a statistically significant difference in path lengths between the conditions in the opposite direction that we expected. The longest path length possible was 15 and the shortest was 1. Such a small difference between means (of .03), although statistically significant, because of the unequal sample sizes, is likely to reflect the path length’s sensitivity to sample size rather than reflecting any meaningful difference
16	Lastly, I’m a bit confused about the concept of identity entrepreneurship in the title. The term was coined by Reicher and Haslam and refers to a leadership process. It might be that it is entailed in dyadic interactions as well but it’s not clear how/ where you see it here and how it was conceptualised/operationalised.	This is a really good point. On reflection we acknowledge that a more appropriate phrasing is ‘identity construction’ or ‘identity creation’. We have amended this throughout.
17 R4	While I like both studies, it is important to note that they tackle quite different research questions. In this regard, playing the devil’s	This is a good point and a similar point was also raised by reviewer three (see comment 15 and the

advocate, I wondered whether the results of study 1 could be interpreted as reflecting some form of “ordinary” conformism with groups? The difference would be that people just conform to the norm consisting in the expression of attitudes by other ingroup members (without considering how this affects their audience) whereas the SIP framework accords a major role to the audience. It would be useful to clearly articulate why the SIP framework offers a better account of these trends a simple conformism account.

response). Both studies do tackle different research questions. Particularly, in study two we assessed identity performance, performative attitude alignment, and attitude-based social identification. It was not possible to directly measure these aspects using large-scale twitter data tracking attitudes about the Russo-Ukraine crisis as it was not possible to directly measure individuals’ group identity. Nonetheless, study one shows a pattern of behaviour as it exists ‘in the wild’. People tend to share similar attitudes to those closest to them in social networks. Importantly, any expression of attitudes in the online context (i.e., expressing a hashtag on Twitter) is done with the knowledge that one is communicating with an audience (Marwick & boyd, 2010; boyd, 2006; Ellison et al., 2006; Litt & Hargitai, 2016). Mere conformism does not require this public expression of attitudes, thus, attitudes expressed online are purposeful acts. Similarly, mere referent informational influence does not require expression of identity but internalisation of norms and information. The social identity performance model, however, does provide an account as to why people would be motivated to publicly express their attitudes (to consolidate and motivate identity). Indeed it could be argued that people are even more conscious when expressing an attitude online because online attitude expressions are more permanent (boyd, 2010; Quinn, 2014) and because they have the potential to reach larger audiences (Bliuc et al., 2020).

We have added a new section in the introduction to make this logic clearer and thank the reviewer for this suggestion.

Page 8

“Qualitative research has found that social identity can be actively [re]constructed and [re]defined on social media and that social media can be used as a tool to display and express one’s social identity (Karizat et al., 2021; Jenkins et al., 2016; Yau et al., 2019; Talbot et al., 2022). Furthermore, previous research has found that, when expressing attitudes online, people are aware of and attentive to their audience (Marwick & boyd, 2010; boyd, 2006; Ellison et al., 2006). That is, the influence of the audience on communication translates to the online context (Mishra et al., 2017). Although people can comprehend the limitless nature of the social media audience, people nonetheless tend to behave as if the audience is more confined and

		imagine a particular audience when posting on social media (Marwick & Boyd, 2010; Litt & Hargitai, 2016). In line with the social identity approach (19, 20), in the online context (as in the offline context) people should be motivated to be similar to their online ingroups (20). Posting an attitude online is a social act that goes beyond any similarity motivations however. That is, it is possible to be referentially influenced by one's online ingroups without needing to post anything online. The social act of sharing one's attitude online goes further than mere referential informational influence. In line with the social identity performance model (2) we propose that people are motivated to publicly express attitudes online in order to consolidate and mobilise their identity (2; 60). People may look to others close to them in a social network and share similar attitudes to them because they are perceived to share an online social identity and because people are motivated to consolidate and mobilise this identity. In line with our main proposal (that interaction with people perceived as ingroup members will lead to alignment on attitudes) we also explore whether online communities tend to express similar attitudes as those close to them in the social network. The purpose of this exploration is to determine whether patterns of attitudinal expression in the large scale online social media context suggest that people could be performatively express attitudes similar to their online groups to consolidate their online identities. We do not directly test the motivations behind these patterns, we simply explore and observe large-scale patterns of attitude expression within online communities on Twitter."
18 R4	The results of study 1 could be interpreted either in "inductive" (i.e., communities form bottom-up based on the expression of common views – through an aggregation mechanism – homophily driving group formation) or "deductive" (i.e., people express views that are consistent with those expressed by other members of the communities because they identify with these – this would be bottom-up). The paper seems to make a case for the latter view, which is consistent with self-categorization theory (and the social identity performance approach). But it is not clear to me that the proposed analyses allow to clearly disentangle these two possible explanations. I assume that by doing analyses over time (i.e., showing that the communities pre-exist the use of the tags), it would be possible to more convincingly do so. It may also be possible to respond to the above question.	We agree that the suggested analyses (over time) would have been an interesting way to tease out which of these are at play. Unfortunately, an analysis over time would not be possible as it is not possible to access this information from the Twitter API. The communities are based on followers, which we cannot gather at multiple time points through the Twitter API (which now appears to be unavailable in any case). The communities we see would be the same at any timepoint. We could perhaps see how the counts of hashtags within each communities changes with time, however this would likely reflect

		events happening, and tell us “when” people talked about particular issues more which is a different research question. Nonetheless, we do not think that one exclusively occurs without the other. We think that both are potentially true. Thus, if a person meets a stranger and, through the process of interaction, they learn that they have a congruent attitude they should inductively bottom up foster a sense of identification as a result of their perceived attitude similarity. Indeed this was our manipulation in study one and our manipulation check (page 27) verifies that this is the case. Indeed this is consistent with previous research in this area (see for example O’Reilly et al., 2022). We also agree with the reviewer’s statement that this process can be “deductive” (I.e., people express views that are consistent with those expressed by other members of the communities because they identify with these”. Indeed that is what we find for hypothesis one. When interacting with attitude-based ingroups, people tend to align their attitudes. When interacting with outgroup members they tend to disalign. This suggests that people are motivated to express views that are consistent with those expressed by members of their ingroups. Indeed we argue for the dynamic nature of this relationship – that the latter (ingroup motivated attitude alignment) can strengthen the original inductively formed attitude-based group identity. Both are also consistent with the SCT and SIP.
19 R4	I am not an expert in network analysis. Upon reading the results, I found that I needed some clarification on several aspects. Table 1 and 2 only report crosses to describe the association between hashtags and communities. Why not report actual frequencies? This would allow computing inferential statistics and also apply descriptive methods allowing to map the communities on the hashtags (thinking of factorial correspondence analysis as a possibility).	We have now visualised the frequencies of hashtag use in each community via a balloon graph to visually represent the relative magnitude of each community’s hashtag use. In our original submission there was something wrong with the supplementary materials file meaning that reviewers were unable to access it. In this file we did include a table of frequencies. In this table we only included frequencies for the most popular hashags but the reviewer makes a really good point that we should include all frequencies. We now have updated this table and report frequencies for all hashtags. We include this table in the supplementary materials as we felt that visually it was less clear than the table with the X’s (supplementary Table1) . We are happy to

	It may also be useful to explain in more detail the meaning of the network indicators used in study 2 (i.e., clustering, path length, edges) and how they adequately operationalize the constructs of interest. I had to trust the authors' knowledge of the subject to be convinced here (to avoid cluttering the MS, a more extensive explanation could be provided in the SM or a source easily understandable by non-experts pointed to).	move this to the main text at the editor or reviewer's request. We also ran the inferential statistics suggested by the reviewer although we did not include factorial correspondence visualisations as we feel that the network visualisation and balloon graph already capture much of the information that these provide. However, we do now report the related chi square goodness of fit tests. Page 14 “We compared the distribution of each hashtag across communities statistically, and in each case a significant chi square goodness of fit statistic demonstrated that the distribution of hashtags was not even across communities (Supplementary Analysis 1). Thus, the results depicted in Table 1 differ significantly from a random distribution.” We have added explanations of these terms as well as a citation for readers to do any follow up reading necessary. Page 27 “After creating the networks where the edges represent shared agreement of participants, we computed the clustering coefficient and the average path length. The clustering coefficient is the fraction of closed triangles in the network, if a node is connected to two neighbours, it gives the probability those neighbours are also connected (see Newman, 2010 for further explanation). The average path length represents the mean shortest number of steps between a pair of participants, in social networks this is related to the idea of six degrees of separation which hypothesises the mean number of steps between any two people is six.”
20 R4	With respect to the analysis of study 2, the authors relied on mixed models. More information should be provided on how these models were specified and why (i.e., in terms of random factors especially) without having to read the syntax.	We have edited the text of the results relating to hypotheses two (pg 33) and three (pg 37), providing more details about the model specification for each, and providing a justification for the choice of random effects used to test hypothesis three. For hypothesis two, although this analysis was conducted through the linear mixed model function in SPSS, all effects were fixed effects. No random effects were specified as total congruence and total identification were used. We have edited the text of hypothesis two, referring to this analysis as a linear model, rather than a linear mixed model, and specifying that total congruence and total identification were used.

21 R4	Something that may be worth clarifying regarding study 2 pertains to the role of the quasi-experimental factor. Given that participants were not randomly allocated to ingroup and outgroup (which makes perfect sense here), is it possible that a confounded factor, such as the content of the attitude play a role in some of the results? Thus, if Prolific participants tend to be “pro-Ukraine”, it is likely that pro-Ukraine attitudes are more represented in the experimental ingroup than in the experimental out-group condition (it would be less likely to find two pro-Russia participants in the same dyad than two pro-Ukraine). In this regard, it would be useful to report the “raw” attitudes (I assume the items form a scale) in each group. Further, if this is the case, I wonder to what extent the clustering observed in the ingroup audience condition may be in part due to the ingroup condition being composed of a more homogeneous group in the first place, reflecting a form of polarization. This may be worth discussing.	The use of the control condition helps to tackle the confound of the content of the attitudes and the natural homogeneity of the groups with respect to attitude alignment. The experimental conditions and the control conditions all answer the same attitudes, the key difference is that the experimental condition engage in attitudinal interactions with their dyadic partners whereas the control conditions don't. Thus, if our effects were merely as a result of naturally occurring group homogeneity and attitudinal alignment (rather than being as a result of interaction, identity performance, and audience effects) we should expect equal amounts of attitude alignment in the experimental ingroup and control ingroup conditions. Similarly, we should expect equal amounts of attitude disalignment in the experimental outgroup and control outgroup conditions.
		It is also important to note that the stimulus questions were purposefully worded in such a way that answering in the affirmative or negative did not necessarily convey a ‘pro-Russian stance. For example, being anti-sanctioning Russia does not necessarily mean a ‘pro-Russian’ stance “Many companies have withdrawn from Russia. Some argue that it is necessary to destabilize Russia and undermines Putin's position and control. Others argue that this leaves innocent civilians without employment or access to necessary resources. Do you think that companies should withdraw from Russia?”. This reflected the Twitter discourse where anti-sanctioning rhetoric generally reflected a ‘pro-peace’ outlook.
22 R4	I found it particularly interesting, and consistent with SCT, that the outgroup audience leads to attitude misalignment. I am puzzled however by the finding that such misalignment is not associated with greater identification. Using a SCT framework, one would probably expect that expressing views that differentiate the ingroup from the outgroup (in a very salient intergroup context) should lead to higher identification. This should be discussed.	Misalignment was associated with lower identification with the outgroup (the dyadic partner is an outgroup because they began with attitude congruence). We did not measure identification with ingroup here as they were not interacting with an ingroup. It would be very interesting for future research to assess whether attitude misalignment after outgroup interaction strengthened attitude-based ingroup identification (and indeed as suggested by the reviewer this would be in line with SCT).
23 R4	In this regard, a possibility that should be considered is that, when responding to the identification scale, people are still “performing” their identity rather than expressing a “private” view. If this is the case, downplaying one’s identity in front of a person discovered to hold a very different attitude from one’s own (outgroup) may be a strategy to facilitate interpersonal interactions. In this respect, it would be useful to report whether participants were sure that their responses were not visible to the audience?	At every instance that a participant’s answer was to be shared with their dyadic partner we clearly stated to the participants “your answer will be shared”. This should hopefully have made it clear to participants which answers would be shared and which would not, thus, they should have been clear that their answers to the social identification questions were not visible to the audience.
24 R4	In our paper (Klein et al., 2007), that the authors use as one of the main foundations of this work, we highlight two possible roles of	This was a very interesting point, thank you. We have now added a section to the discussion.

	social identity performance: one is consolidation, the other mobilization. The authors focus on the first function. However, they show that the outcome of this function is actually successful mobilization. It would be interesting in the discussion to discuss this possible paradox and/or question the value of this distinction.	“More speculatively, situations of uncertainty and threat, such as the Russian invasion of Ukraine, may perpetuate attitude propagation. When interacting with an ingroup, we may be motivated towards attitudinal alignment, not only to pursue goals of ingroup inclusion, but also because ingroups have collective power only when they achieve consensus or cohesion on core dimensions. When attitudes become coordinated and collective, they have the power to transform general social opinion on issues (e.g. women’s/ LGBTQ rights movements) as well as ignite collective action and transform macro level societal structures (4; 3). As group cohesion maximises the collective agency and power of a group (610; 3; 7), the strategic alignment of attitudes should be especially pertinent in situations of threat, such as the Russian invasion of Ukraine. Finding that those who engaged in attitudinal interaction aligned their attitudes more than those who privately expressed their attitudes in the context of the Ukraine crisis, may provide preliminary support that people are motivated to align their attitudes to achieve collective power. Indeed, this would reflect the identity mobilisation dimension of the social identity performance model (2). Klein and colleagues explain that, not only can identity performance serve to consolidate identity but also to mobilise it. That is, identity performance can serve to foster ingroup related action. It is difficult to ascertain from the current results whether or not people performatively expressed attitudes to mobilise identity. Of course, this assertion is speculative and future research is needed to probe these effects further.”
25 R4	My name (“Klein”) is misspelled on several occasions as Klien.	We have changed the incorrect versions. We apologise for this mistake and sincerely hope no offence was caused.
26 R4	Figure 3 comes straight from SPSS and could be improved visually.	We assume the reviewer meant figure 4. This was a good point. We have now updated this figure to display individual data points (as per stipulations of this journal) and improved it visually. Page 35

27
R4 Something that I found puzzling is that so many countries are represented in the sample although the native language of the participants was English. Is this not an error?

Although inclusion criteria was any country in geographical Europe, the representation of countries based on participants' open text answers to "please enter your current country of residence" and "If different from the previous entry, please enter your country of birth" were as follows:

Country of residence

- Britain 4
- Great Britain/UK 994
- England 275
- Ireland 15
- Northern Ireland 8
- Scotland 44
- Wales 17
- Poland 2
- Spain 2
- Sweden 1
- Germany 1
- The Netherlands 1
- Other 10

Country of birth

- America/United States 11
- Australia 7
- England 31
- Bosnia and Herzegovina 1
- Botswana 1
- Canada 5
- China 1
- Cyprus 2
- France 1
- Great Britain/United Kingdom 58
- Germany 3
- Ghana 2
- Hong Kong 2
- India 7
- Ireland 9

		Italy 1 Jordan 3 Malaysia 4 Mexico1 New Zealand 2 Nigeria 16 Northern Ireland 1 Oman 1 Pakistan 3 Poland 2 Scotland 5 Singapore 2 Slovenia 1 South Africa 3 Spain 1 Sri Lanka 1 Switzerland 2 The Netherlands 1 Turkey 2 Ukraine 2 Wales 2 Yemen 1 Zambia 1 Zimbabwe 3 Other 3 Note that inclusion criteria on prolific also specified “first language English”. It is certainly possible that some people self-assessed as having good enough English or perhaps that some were not fully truthful about this. However, the majority of participants report living in or being from a country whose majority spoken language is English. Perhaps also some participants may have grown up with first language was English whilst living in a country whose first language is not English (bilinguals). We have updated the wording in the manuscript slightly to account for this Pg 19 “The survey was made available to individuals who self-declared that their whose first language was English”
	Social Network Review	
28 R5	#1 Network boundary Perhaps the first step in collecting network data is a determine the boundary. That is, who is included and not included in the network. The inclusion criteria in the current manuscript is “English containing hashtags relating to the Russia/Ukraine war, starting from the beginning of 206 the invasion (24/02/22) and ending the day we commenced gathering data (28/06/22)”. A couple of questions.	

What was the full list of hashtags? I tried to open the Supplementary Materials document on the OSF page, but it is just a blank document. Perhaps this is just a problem with OSF? Second, what is the rationale for including the specific hashtags? What was the process of determining which to include and which not to include? For instance, many of the hashtags express a valence on the war (e.g., #stoprussia), some are incredibly general (e.g., #peace), and some strike me as irrelevant (e.g., #ethiopia— am I missing something here?). Finally, isn't a bit surprising that the data only captured 8,149 users? This was one of the biggest stories of 2022. It strikes a red flag regarding validity if the inclusion criteria only was able to extract that amount of users. Perhaps many people tweeting about the war just didn't use hashtags?	The supplementary materials issue has now been resolved and the link in the OSF page now functions. We did not make our method clear enough and have now updated it. We used a list of hashtags related to the Russo-Ukraine crisis as our inclusion criteria. We then narrowed this sample down by choosing a subset of hashtags and including everyone who had tweeted or retweeted one of these. We then looked at all tweets from these users. This process is what resulted in such a small portion of the original data being chosen. We also removed any accounts where we couldn't access a user's followers (due to the account being deleted, banned or otherwise made inaccessible through the Twitter API.) Updates Page 12: “Participants and Procedure We searched twitter to identify popular hashtags related to the ongoing invasion of the Ukraine, as well as hashtags that were ‘trending’ at the time (see Supplementary Note 2 for full list of hashtags), attempting to identify as many dimensions of the online discourse as possible. Participants were included in the dataset if they had tweeted any of the hashtags on the hashtag list between February and June 2022 (from Russia’s invasion of Ukraine and for five months after this). This yielded 5,000,000 tweets from 1,000,000 users. We then examined the frequencies of hashtag expressions for each tweet and chose six hashtags that had high tweet or retweet frequencies. Our final sample consisted of a subset of users who had tweeted one of these hashtags (Supplementary Note 4). We then created a dataset of all tweets in the original dataset from users who were in our final sample. Next, we gathered the follower list for each of these users, discarding the data for any user whose account information was unavailable through the Twitter API (e.g., restricted, banned, or deleted accounts). We also discarded any user in the follower lists for whom we had no tweets. This left us with 8,149 users. We then built a followership network of these users, where users were linked to every user they were followed by, if that user had also tweeted or retweeted one of the hashtags.“
29 R5	#2 No rationale for undirected network

	Twitter is an undirected network. That is, just because you follow me, doesn't mean I follow you back (unlike Facebook friends). As such, I see no point in analyzing the network as undirected because we are then artificially creating networks. This especially true because further analysis is done on the network as directed. I highly advocate removing this analysis because it could provide misleading interpretations of the data.	The authors agree with this comment and have removed the undirected network section.
30 R5	#3 Why largest component Unless I am mistaken, there is no assumption in the Infomap community detection algorithm that the network be a single component. As such, what is the rationale for only applying the algorithm to largest connected component? Are we not losing information on the more peripheral regions of the network? This unanalyzed region seems to be pretty big too if only 3812 nodes were analyzed.	We agree and we have now applied the algorithm to the entire network (instead of applying it solely to the largest connected component). Note that the nodes that are further analysed (relating to their hashtag use) include those who belong to larger groups (groups larger than 100). Our figure have changed slightly with this update (see Figure 1) but the overall results remain consistent.
31 R5	#4 Footnote 2 Four hashtags were removed because they were “not that informative”. What makes a hashtag informative? What is the criteria? Why are the ones included in the tables informative and others not?	We were attempting to assess whether the communities differed in their use of hashtags relating to the Russo-Ukraine crisis. The four hashtags #Ukraine, #Ukrainian, #Russia and #Russian were removed because these hashtags were too general to be informative beyond merely indicating that people were discussing the Russo-Ukraine crisis. That is, they did not give further information about the specific nature of the discussion. We have included the following statement in the manuscript to clarify Pg 11 “We excluded several hashtags such as #Ukraine, #Russia (Supplementary Note 2) because these were hashtags used by all communities and therefore not that informative of differences between the communities. That is, these hashtags did little more than signify that people were discussing the Russo-Ukraine crisis at large but were not informative of any qualitative nuances in the discourse.”
32 R5	#5 Lack of statistical evidence The manuscript writes that “These results demonstrate that individuals tend to publicly express hashtags that others in their follower communities also express.” However, what information from Table 2 gives the reader evidence that this is a correct interpretation? For instance, if the null hypothesis was true (communities don't use the same hashtags), what would the table look like? This data seems ripe for some sort of statistical analysis. For instance, in one column we have a categorical variable (community membership) and the other is a multivariate continuous variable	This was really good advice, thanks to the reviewer. We have added some inferential analyses to demonstrate further our interpretation. If the null hypothesis was true we would not expect the X's in the table to differ between the communities. “We compared the distribution of each hashtag across communities statistically, and in each case a significant chi square goodness of fit statistic

	(number of times a user used a particular hashtag). Essentially, we want to see if hashtag use has an association with community membership. Perhaps analyzing on the individual level makes sense. That is, use community membership as a categorical variable to predict each hashtag use as a continuous variable. An ANOVA for each hashtag should do the trick, no?	demonstrated that the distribution of hashtags was not even across communities and hashtag use characterises these communities (Supplementary Analysis 1). This demonstrated that the results depicted in Table 1 differ significantly from a random distribution. We also ran individual ANOVA's to assess the frequency of individuals' hashtag use across all communities. 20 out of 21 showed statistically significant differences (Supplementary Analyses 2). Thus, followership communities on twitter are characterised and clearly distinguished by the combination of hashtags that they publicly express. Taken together, these results all suggest that groups of followers tend to use similar hashtags depending on what "group/community" they are part of. "
	"Minor points"	
33 R2	1. 1.204 You should say "data were" (data is plural)	Thanks, we have amended this error.
34 R2	2. For Figure 6, I would separate the title of the figure from a note that then talks readers through this.	We have done this.
35 R2	3. In Table 5 I wouldn't say "Attitudes asked" but would refer to these as "Stimulus questions"	We have done this.
R2	Positive Comments	
	This paper investigates a very interesting question — namely how people align their attitudes with those of ingroup members and what role identity leadership plays in this process. This pertains to a burgeoning area of interest in a range of spheres (not just social and political psychology, but also organizational and health psychology). In line with other recent work in this vein, this paper offers clear insights into these issues that have the capacity to ingrate and extend a range of previous findings and insights. Even though the theoretical arguments that inform this research are quite complex, the logic and rationale for the paper was generally very clear, as was the overall case that it presented. Moreover, I think the paper will be accessible to Communications Psychology readers and of considerable interest to them – as well as to a broader readership interested in the issues of leadership, attitudes and collective behaviour that it explores. It is also worth noting too that while the arguments presented here make very good sense from the perspective of a social identity approach (e.g., Haslam et al., 2011), they are likely to be quite novel to many readers, and accordingly, I think the paper should have considerable value in helping to signal current developments in the field. So while a little bit more work is needed to really underline and sharpen the paper's contribution (see below), I am generally supportive of its publication.	We very much appreciate the reviewers' positive comments.

R3	Thank you for giving me the opportunity to review the paper “Strategic attitudes as identity performance and identity entrepreneurship in interaction”. There’s much to like about this paper; it is well written and informative and looks at an interesting topic, namely how identity and attitudes co-emerge in interaction. The work is based on Social Identity Theories, for example, the SIDE model and work by Haslam, Reicher and others.	
R4	This paper addresses the concept of social identity performance in two studies. In the first study, the authors show that, in the context of the Russia-Ukraine war, online communities distinct hashtags. The second study, using a quasi-experimental method, examines how attitude alignment as a function of the group membership of the audience & and awareness thereof and how this influences identification. It convincingly shows that such alignment helps consolidate identity in the presence of an ingroup audience. I very much appreciated reading this paper for a variety of reasons:  - It capitalizes on a major political event to study group behavior “in the wild” but combines this with a much more controlled experimental study. - This is a strong contribution to one of the dominant approaches to group behavior (the social identity approach) and allows extending it to new perspectives by showing not only how identity expressions are a consequence of the salience of group identities but serve to construct group identities. This idea of “identity construction” is often used in qualitative work and social constructivist perspectives (in sociology, anthropology, etc.). Such has many merits, but often suffers from a difficulty to model psychological processes (with the added risk of subjectivity). Study 2 especially, allows to empirically establish this process of identity construction using a novel and innovative experimental method. This is a major contribution. - The authors use methods that have been very seldom used in the social identity literature to tackle issues of central interest to them. The use of network analysis to identify communities and the clustering of attitudes is especially worthy of interest in this regard. - The findings have wide theoretical and societal implications for how large group of people may come to synchronize attitudes. Specifically, public discourse on mass opinion formation often uses an “epidemiological” metaphor, suggesting that attitudes are spread like viruses without considering how the expression of such attitudes serves distinct needs associated with specific psychological processes. By engaging such processes, the paper offers a fresh and innovative perspective on such mass opinion formation. - I appreciated that study 2 was preregistered and that the authors made their material and analyses transparent. 	

28th Sep 23

Dear Ms O'Reilly,

We profusely apologise for the delay in processing your manuscript titled "Strategic attitude expressions as identity performance and identity creation in interaction" and thank you for your patience.

To explain what caused the delay: one of the previous reviewers could not return an evaluation of your manuscript. This was not immediately known to the editorial team when we send your work back to review; because they were the only reviewer ideally placed to assess some of the methodological aspects of your work, we were unable to proceed to a decision on your manuscript without replacing them. Unfortunately, soliciting a report from a new expert in this domain took much longer than anticipated. Your manuscript has now been seen by 4 reviewers, Reviewers #1, #2, and #3 from the first round, and Reviewer #5 who provides the same expertise as the previous Reviewer #4.

Their comments are included at the end of this letter.

You will see that Reviewers #1, #2 and #3 appreciate the changes you undertook in revision and Reviewers #1 and #3 are fully supportive of your work. In light of the positive evaluation by Reviewer #1 and #3, and the remaining important concerns voiced by the other reviewers, we ask you to undertake one final set of revisions before we make a decision on publication.

Reviewer #2 has some remaining concerns which we ask you to address through careful discussion of potential limitations of the approach. Like Reviewer #2, Reviewer #5 also has concerns about the support your Study 1 offers for the social identity performance model. To respond to these concerns, we ask you to undertake an additional analysis of the data.

We therefore invite you to revise and resubmit your manuscript, along with a point-by-point response to the reviewers. Please highlight all changes in the manuscript text file.

Please note that your revised manuscript must comply with our formatting and reporting requirements, which are summarized on the following checklist: <https://www.nature.com/documents/commspsychol-style-formatting-checklist-article-rr.pdf> Communications Psychology formatting checklist and also in our style and formatting guide <https://www.nature.com/documents/commspsychol-style-formatting-guide-accept.pdf> Communications Psychology formatting guide .

Please use the following link to submit your revised manuscript, point-by-point response to the referees' comments (which should be in a separate document to any cover letter) and the completed checklist:

[link redacted]

Please do not hesitate to contact me if you have any questions or would like to discuss these revisions further. Again, we are sorry for the unconscionable delay and thank you for your patience.

We look forward to seeing the revised manuscript.

Best regards,

Antonia Eisenkoeck

Antonia Eisenkoeck
Senior Editor
Communications Psychology

EDITORIAL POLICIES AND FORMATTING

Editorial Policy: [Policy requirements](https://www.nature.com/documents/nr-editorial-policy-checklist.pdf) (Download the link to your computer as a PDF.)

*** TRANSPARENT PEER REVIEW:** Communications Psychology uses a transparent peer review system. This means that we publish the editorial decision letters including Reviewers' comments to the authors and the author rebuttal letters online as a supplementary peer review file. However, on author request, confidential information and data can be removed from the published reviewer reports and rebuttal letters prior to publication. If your manuscript has been previously reviewed at another journal, those Reviewers' comments would not form part of the published peer review file.

*** CODE AVAILABILITY:** All Communications Psychology manuscripts must include a section titled "Code Availability" at the end of the methods section. In the event of publication, we require that

the custom analysis code supporting your conclusions is made available in a publicly accessible repository; at publication, we ask you to choose a repository that provides a DOI for the code; the link to the repository and the DOI will need to be included in the Code Availability statement. Publication as Supplementary Information will not suffice. We ask you to prepare code at this stage, to avoid delays later on in the process.

*** DATA AVAILABILITY:**

All Communications Psychology manuscripts must include a section titled "Data Availability" at the end of the Methods section or main text (if no Methods). More information on this policy, is available at <http://www.nature.com/authors/policies/data/data-availability-statements-data-citations.pdf>.

At a minimum the Data availability statement must explain how the data can be obtained and whether there are any restrictions on data sharing. Communications Psychology strongly endorses open sharing of data. If you do make your data openly available, please include in the statement:

We recommend submitting the data to discipline-specific, community-recognized repositories, where possible and a list of recommended repositories is provided at <http://www.nature.com/sdata/policies/repositories>.

If a community resource is unavailable, data can be submitted to generalist repositories such as [figshare](https://figshare.com/) or [Dryad Digital Repository](http://datadryad.org/). Please provide a unique identifier for the data (for example a DOI or a permanent URL) in the data availability statement, if possible. If the repository does not provide identifiers, we encourage authors to supply the search terms that will return the data. For data that have been obtained from publicly available sources, please provide a URL and the specific data product name in the data availability statement. Data with a DOI should be further cited in the methods reference section.

REVIEWERS' EXPERTISE:

Reviewer #1: effect of group on attitudes, social identity
Reviewer #2: effect of group on attitudes, social identity
Reviewer #3: effect of group on attitudes, social identity
Reviewer #5: social network analysis

REVIEWERS' COMMENTS:

Reviewer #1 (Remarks to the Author):

I was positively disposed to this paper upon reviewing it previously, and think it has been significantly improved through attention to the points that the reviewers raised with the first iteration of the manuscript.

The key point for me is that it is still a very good paper — provocative and insightful in equal measure. Indeed, I think that the changes that the authors have made do a good job of teasing out the distinctive contribution of the paper to the field.

Accordingly, I am happy to recommend that it be published, and look forward to being able to cite it in future.

Reviewer #2 (Remarks to the Author):

Thank you for the rebuttal letter and your changes in the manuscript. Many of my original questions were well addressed.

As you mentioned, the other reviewers liked Study 1 and I'm happy with that. However, I personally don't think that you can make strong claims beyond the mere fact that people in a shared network share hashtags and I'm not sure whether your claim that " In line with the social identity performance model (2) we propose that people are motivated to publicly express attitudes online in order to consolidate and mobilise their identity (2; 60). People may look to others close to them in a social network and share similar attitudes to them because they are perceived to share an online social identity and because people are motivated to consolidate and mobilise this identity." can be deduced from the observed patterns of different hashtag use in different networks. You acknowledge that in the next section by saying " we simply explore and observe large-scale patterns of attitude expression within online communities on Twitter." This last sentence I agree with, although again, we don't even know whether the use of a specific hashtag is an expression of an attitude or just a away of 'pinning' and 'connecting' with others.

There are a number of typos and odd full-stops/ spaces, missing spaces; font changes sometimes (p27) throughout; please carefully edit your manuscript before the final submission

Reviewer #3 (Remarks to the Author):

I read the rebuttal letter and the new version of the MS. I was rev 4 for the initial submission and found that the authors did a good job answering my and the other reviewers' comments. I however found quite a few minor errors / weaknesses in the MS. I am appending a PDF with suggested edits.

Olivier Klein

[Editorial comment: To avoid any doubt, please note that there seems to be a small mistake here, Reviewer #3 was Reviewer #3 in the previous round as well]

Reviewer #4 (Remarks to the Author):

Study I

Focusing on the context of the Russo-Ukraine war, Study I conducts a descriptive network analysis using tweet data. This analysis aims to determine whether the patterns of attitudinal expression on Twitter align with the attitudes expressed by those in close online followership networks.

The aggregate-level analysis is generally sound. The strategy for sampling relevant tweets, users, and hashtags appears logical. The authors' choice of community detection algorithm is compelling, as are the subsequent statistical analyses.

However, a significant concern is the absence of micro-level evidence. The social identity performance model suggests that individuals are inclined to publicly express attitudes that resonate with their in-group members' views. The most effective method to verify this would be to examine individual expressions and compare them to the expressions of those they follow. Unless I'm mistaken, the authors' dataset should permit such an analysis. Without justifying why an aggregate-level analysis is preferable or adequate compared to individual-level analysis, I believe this section of the paper does not offer a thorough examination of the model.

Some minor points: Several sentences need revision.

- It's unclear what the authors mean by "both algorithms" (line 304) and "both Figures" (line 326).
- There's a typographical error: "Figure"s" 1" (line 328).
- The network visualization needs more explanation (e.g., with a caption).

Study II

To scrutinize their argument, the authors proceed to conduct a RCT. Specifically, the network analytic approach the authors take is quite novel and well-designed. With little deviation from the pre-registration, the authors employ two different measurement strategies: one based on path-length, and the other one based on clustering coefficient. Also, the authors' interpretation of the results and discussion of limitations (stemming from unequal sample sizes on p. 33) sound persuasive.

27 November 2023

2	Reviewer 2	by saying" " we simply explore and observe large-scale patterns of attitude expression within online communities on Twitter." This last	This is a really good point. We did not intend to make such strong claims and agree that our study does not provide sufficient evidence to support the claim made in the section quoted by the reviewer. It was not our intention to imply that we were testing this claim and can see that our wording required changing here. We have updated our phrasing and we hope that it is now clear that we are simply exploring and observing. “People may be motivated to publicly express attitudes online in order to consolidate and mobilise their identity (Klein et al., 2007; Moojiman et al., 2018). People may look to others close to them in a social network and share similar attitudes to them because they are perceived to share an online social identity and because people are motivated to consolidate and mobilise this identity. In line with our main proposal (that interaction with people perceived as ingroup members will lead to alignment on attitudes) we thus also explore whether people in online communities tend to express similar attitudes as those close to them in the social network. The purpose of this exploration is to determine whether patterns of attitudinal expression in the large scale online social media context suggest that people could be performatively expressing attitudes similar to their online groups to consolidate their online identities. We do not directly test the motivations behind these patterns, we simply explore and observe large-scale patterns of attitude expression within online communities on Twitter.” We have acknowledged this in the limitations. We have also discussed the point made by the reviewer that hashtags are not always attitude expressions.
----------	-------------------	--	---

			Pg 52 “Theoretically, we expect that, in line with the social identity performance model (Klein et al., 2007), people can be motivated to publicly express attitudes online in order to consolidate and mobilise their identity. We expect that people can look to others close to them in a social network and share similar attitudes to them because they are perceived to share an online social identity and because people are motivated to consolidate and mobilise this identity. It is important to note that, while the online patterns of attitude sharing we observed in the Twitter study are in line with these theoretical suggestions, we have not ruled out alternative explanations as this was merely an exploratory observational study of large scale attitude sharing patterns.” Pg 52 “In the current paper, we conceptualise ”
3	Reviewer 4		We thank the reviewer for their positive comments. Thanks to the reviewer for this great suggestion. We have now undertaken additional analysis to examine individual expressions as the reviewer suggests. We have obtained a sentiment score of each user from their tweets in the data and compared the average sentiment across communities found in the data. To do this, we ran an ANOVA with Community as the independent variable and sentiment score as the dependent variable. Results demonstrate

			that, on average, variation in sentiment is larger across communities than it is within communities. Pg 18 “On top of the aggregate level analysis above, we also conducted individual level analysis. Using AFINN sentiment lexicon (specifically designed for microblogs; Nielsen, 2011), we obtained a sentiment score from the tweets of each user and compared the average sentiment across the three largest communities. A one-way ANOVA (see supplementary figure 1, supplementary table 2) demonstrated that the average sentiment for individuals in the followership network was significantly different across communities (community zero: $M = -1.35, SD = 1.63$; community one: $M = -0.51, SD = 1.60$; community two: $M = -0.85, SD = 1.68$), $F(2, 4290) = 78.86, p < .001, \eta_p^2 = 0.035$. These results demonstrate that, on average, variation in sentiment is larger across communities than it is within communities.”
	Minor points		
4	Reviewer 2		Many thanks for spotting them.
5	Reviewer 4		This was a mistake leftover from a previous revision of the paper which we have now amended.
6	Reviewer 4		We have amended this. Thanks for spotting it.
7	Reviewer 4		We have updated the following figure label “Bipartite attitude alignment for each condition. The nodes are coloured based on their identification score, (red = low score, green = medium score, blue = high score) and an edge is created when the participants agree on X or more of the 11 items. The network for the experimental ingroup condition is more dense with a lower mean path length. This suggests that there is greater attitude alignment in the experimental ingroup condition.”

8	Reviewer 3		We thank the reviewer for their positive comments. We have now amended all of the minor errors spotted by this reviewer.
	Positive comments		
9	Reviewer 1	The key point for me is that it is still a very good paper —	Many thanks for the positive comments, we very much appreciate the supportive feedback.
10	Reviewer 4		Many thanks for the positive comments, we very much appreciate the supportive feedback.

9th Jan 24

Dear Dr O'Reilly,

Your manuscript titled "Strategic attitude expressions as identity performance and identity creation in interaction" has now been seen by our reviewers, whose comments appear below. In light of their advice I am delighted to say that we are happy, in principle, to publish a suitably revised version in Communications Psychology under the open access CC BY license (Creative Commons Attribution v4.0 International License).

We therefore invite you to revise your paper one last time to address the remaining concerns of our reviewers and a list of editorial requests. At the same time we ask that you edit your manuscript to comply with our format requirements and to maximise the accessibility and therefore the impact of your work.

EDITORIAL REQUESTS:

Importantly, you are requested to revise the presentation of your manuscript to adhere to journal policies. In short, you need to shorten the main text; aligning the sections with the required style will give you the opportunity to do so (see Editorial Request Table for details). We strongly discourage overlap between introduction and discussion (discussion points should be only in the discussion) and any discussion must be removed from the results reporting. This will free up space for something that is much more important - transparent reporting of methods and results. All key preregistered analyses (and their results) must be in the main article, not in the supplement. Likewise, the analysis you included as part of the review process must be reported in the Methods and Results (labelled 'Non-registered analysis').

Only ancillary analyses can be in the Supplementary Information file.

SUBMISSION INFORMATION:

OPEN ACCESS:

Communications Psychology is a fully open access journal. Articles are made freely accessible on publication under a [CC BY license](http://creativecommons.org/licenses/by/4.0) (Creative Commons Attribution 4.0 International License). This license allows maximum dissemination and re-use of open access materials and is preferred by many research funding

bodies.

For further information about article processing charges, open access funding, and advice and support from Nature Research, please visit <https://www.nature.com/commspsychol/article-processing-charges>

At acceptance, you will be provided with instructions for completing this CC BY license on behalf of all authors. This grants us the necessary permissions to publish your paper. Additionally, you will be asked to declare that all required third party permissions have been obtained, and to provide billing information in order to pay the article-processing charge (APC).

* **DATA AVAILABILITY:**

[link redacted]

Best regards,

Antonia Eisenkoeck

Antonia Eisenkoeck
Senior Editor
Communications Psychology

REVIEWERS' EXPERTISE:

Reviewer #1
Reviewer #2

REVIEWERS' COMMENTS:

Reviewer #2 (Remarks to the Author):

Thank you very much for considering the points I raised; I'm happy with the revision and can recommend accepting this paper. This is an interesting paper and I thank you for the constructive revision process.

Reviewer #5 (Remarks to the Author):

I've just reviewed the responses from the authors (in the rebuttal letter). I believe that the concerns I raised have properly been handled and have no further concerns.